# PACEAttention: Principled and Adaptive Feature Compression-Expansion Grounded in the Geometry of MCR$^2$

Xiaojie Yu [1]   Haibo Zhang [2]   Jeremiah D. Deng [1]   Lizhi Peng [3 4]

## Abstract

The maximal coding rate reduction (MCR$^2$) objective is proposed for learning low-dimensional subspace representations and for principled deep model design, where layer structures are derived by unrolling its optimization steps. However, existing methods motivated by this objective do not fully adhere to design principles implied by the MCR$^2$ gradient, which weakens the principled and interpretable foundations of the resulting models. In this work, we introduce PACEAttention, a novel principled attention mechanism inspired by the *geometric insight* of MCR$^2$, whose gradient-based updates move features along directions shaped by the underlying low-dimensional feature structure. Our method captures this structure by leveraging randomization to guide feature updates. This principled construction enables the resulting PACENet to exhibit enhanced interpretability, with different heads attending to distinct image regions and capturing *fine-grained* structures under simple supervised training. Experiments demonstrate that PACEAttention achieves superior performance and more stable scalability than previous principled modules while remaining low complexity. Code is available at this https URL.

## 1. Introduction

Despite their empirical success, Transformer-based (Vaswani et al., 2017) architectures are often regarded as black-box models. While their attention mechanisms define how features are updated across layers, these updates are not derived from an explicit global objective that governs representation geometry. In particular, feature updates in Transformers are guided by softmax-normalized similarity scores between queries and keys, a design choice that is effective but difficult to interpret from an optimization or geometric perspective. This raises a fundamental question: *Can feature updates be derived from an explicit objective that governs both representation geometry and layer-wise transformations?* One principled answer is to learn low-dimensional, class-specific subspace representations through an explicit objective. By unrolling the optimization steps of this objective, the gradients of the objective provide explicit update rules, which directly determine the structure and operations of each layer in the resulting deep network. Under this formulation, *feature updates are guided by the intrinsic low-dimensional subspace structure of representations at each layer*, rather than relying solely on pairwise similarity, as in standard attention mechanisms (Chan et al., 2022; Yu et al., 2023; Wu et al., 2025). This unrolled optimization perspective naturally yields objective-driven, white-box layer constructions, in which representation updates admit clear geometric and optimization interpretations.

ReduNet (Chan et al., 2022) unifies these ideas by explicitly constructing each layer as a white-box update obtained by unrolling gradient ascent steps of the *maximal coding rate reduction* (MCR$^2$) objective, which performs compression for intra-class compactness and expansion for inter-class separation. Furthermore, two variants of ReduNet were introduced to reduce computational complexity, attempting to retain the original principle for architecture design. Coding-RATE Transformer (CRATE) approximates a gradient descent step on a modified compression term of the MCR$^2$ objective, yielding a multi-head subspace self-attention (MSSA) operator (Yu et al., 2023). Unlike standard Transformer self-attention, MSSA sets the query, key, and value to be identical. Token Statistics Transformer (ToST) further reduces complexity by optimizing an upper bound of the modified compression term that depends only on the diagonal of the feature correlation matrix, yielding a linear-time attention operator called token statistics self-attention (TSSA) (Wu et al., 2025).

However, the approximation and simplification adopted by

---

[1]School of Computing, University of Otago, Dunedin, New Zealand. [2]School of Computer Science and Engineering, University of New South Wales, Sydney, Australia. [3]Quancheng Laboratory, Jinan, China. [4]School of Information Science and Engineering, University of Jinan, Jinan, China. Correspondence to: Haibo Zhang <haibo.zhang@unsw.edu.au>, Jeremiah D. Deng <jeremiah.deng@otago.ac.nz>.

*Proceedings of the 43$^{rd}$ International Conference on Machine Learning*, Seoul, South Korea. PMLR 306, 2026. Copyright 2026 by the author(s).

these approaches prevent them from fully realizing the principled operations implied by the gradient of the $\text{MCR}^2$, which requires feature updates guided by the *intrinsic low-dimensional subspace structure* of representations. Specifically, it has been shown in (Hu et al., 2024) that MSSA may deviate from its objective and cause the compression term to increase (rather than decrease as expected). TSSA has limited ability to capture fine-grained intrinsic structure, as it assumes a simple diagonal feature correlation matrix and does not account for correlations across dimensions. In addition, ReduNet and its variants do not have a self-adaptive mechanism to balance expansion and compression across layers, while recent studies suggest that emphasizing expansion in the early layers may enhance inter-class feature separation and facilitate intra-class feature compression in deeper layer (Yu et al., 2024; Xu et al., 2025; Alain & Bengio, 2017; Masarczyk et al., 2023; Wang et al., 2025). Moreover, MSSA updates features based on the full feature correlation matrix, resulting in a quadratic computational complexity. Overall, these limitations can compromise both the interpretability and performance of constructed models.

In this work, we revisit $\text{MCR}^2$ from a geometric perspective and show that principled feature expansion and compression can be guided by the intrinsic low-dimensional structure induced by column space representations. Motivated by this, we propose **PACEAttention**, a novel attention mechanism that offers the following key advantages: (1) **Principled expansion & compression**: PACEAttention achieves principled feature updates by leveraging representative low-dimensional subspaces that align with the geometric behavior of the $\text{MCR}^2$ gradient. (2) **Self-adaptivity**: We introduce two learnable weights for expansion and compression, enabling the model to adaptively balance their contributions at each layer. (3) **Better interpretability**: We show that PACENet, the network model based on PACEattention modules, display better interpretability. In particular, different heads within the compression modules attend to distinct fine-grained image regions, yielding diverse and interpretable features. Moreover, self-adaptivity offers a macro-level view of how the model coordinates expansion and compression across layers, with early layers emphasizing expansion when representative dimensions are limited.

PACEAttention achieves *linear complexity* in both time and memory, scaling as $\mathcal{O}(n)$ ($n$ is the number of tokens). Experiments show that PACENet achieves superior performance to CRATE with lower complexity and parameter costs. In particular, despite its linear complexity and only 7.2M parameters, PACENet-S achieves 79.0% top-1 accuracy on ImageNet ReaL, surpassing CRATE-L (77.6M parameters, quadratic complexity) at 77.4%. Moreover, PACEAttention exhibits superior scalability compared to TSSA, whereas the ToST variant constructed solely from the TSSA module struggles to scale, with training becoming unstable and terminating due to numerical errors.

## 2. Background and Related Work

**Notation.** For a vector $\boldsymbol{v} \in \mathbb{R}^n$, let $\text{Diag}(\boldsymbol{v}) \in \mathbb{R}^{n \times n}$ be the diagonal matrix with $\text{Diag}(\boldsymbol{v})_{ii} = \boldsymbol{v}_i$. Let $\boldsymbol{I}$ be the identity matrix. For a positive integer $n$, let $[n] = \{1, 2, \ldots, n\}$. We denote by $\mathbf{1}$ the vector of all ones. We denote by $\text{PSD}(n) \subseteq \mathbb{R}^{n \times n}$ the set of $n \times n$ positive semi-definite (PSD) matrices. For $\boldsymbol{M} \in \text{PSD}(n)$ and $i \in [n]$, we denote by $\lambda_i(\boldsymbol{M})$ the $i$-th largest eigenvalue of $\boldsymbol{M}$.

Raw data such as images are typically first tokenized into vectors $\boldsymbol{X} = [\boldsymbol{x}_1, \ldots, \boldsymbol{x}_n] \in \mathbb{R}^{D \times n}$ for training, where each vector $\boldsymbol{x}_i \in \mathbb{R}^D, i \in [n]$ represents a local part of the data. These tokens often belong to different semantic categories, and conventional deep models mainly aim to learn task-specific representations for prediction. In contrast, ReduNet introduces the $\text{MCR}^2$ objective, which instead focuses on learning task-agnostic subspace representations $\boldsymbol{Z} = [\boldsymbol{z}_1, \ldots, \boldsymbol{z}_n] \in \mathbb{R}^{d \times n}$ from tokens $\boldsymbol{X}$ (Chan et al., 2022). Specifically, tokens of an image are assumed to belong to $K$ groups, and $\text{MCR}^2$ aims to learn token features that are compact within each group while being well separated across $K$ groups. Let $\boldsymbol{\Pi} = [\boldsymbol{\pi}_1, \ldots, \boldsymbol{\pi}_K] \in \mathbb{R}^{n \times K}$ be a membership matrix, where the $k$-th column $\boldsymbol{\pi}_k \in \mathbb{R}^n$ contains the soft assignments of all $n$ tokens to the $k$-th group. Obviously, the assignments satisfy $\sum_{j \in [K]} \boldsymbol{\Pi}_{ij} = 1, \forall i \in [n]$. Let $\epsilon > 0$ and $n_k = \langle \boldsymbol{\pi}_k, \mathbf{1} \rangle$ for each $k \in [K]$. The $\text{MCR}^2$ is defined as follows:

$$
\begin{aligned}
\Delta R(\boldsymbol{Z}, \boldsymbol{\Pi}) &= R(\boldsymbol{Z}) - R_c(\boldsymbol{Z}, \boldsymbol{\Pi}) \\
&= \frac{1}{2} \log \det(\boldsymbol{I} + \frac{d}{n\epsilon^2} \boldsymbol{Z} \boldsymbol{Z}^T) \\
&\quad - \frac{1}{2} \sum_{k=1}^{K} \frac{n_k}{n} \log \det(\boldsymbol{I} + \frac{d}{n_k \epsilon^2} \boldsymbol{Z} \text{Diag}(\boldsymbol{\pi}_k) \boldsymbol{Z}^T)
\end{aligned}
\tag{1}
$$

Here, the expansion term $R(\boldsymbol{Z})$ measures the volume of the overall feature space, promoting inter-class feature separation. In contrast, the compression term $R_c(\boldsymbol{Z}, \boldsymbol{\Pi})$ measures the sum of volumes of class-wise feature subspaces encoded by $\boldsymbol{\Pi}$, and minimizing it encourages intra-class compactness. To maximize Eq. (1), the operators in each layer of ReduNet, roughly in the form of $(\boldsymbol{I} + \boldsymbol{Z}\boldsymbol{Z}^T)^{-1}$, are derived from the gradient ascent step of $\text{MCR}^2$, and perform expansion and compression with equal weights. While ReduNet offers a principled design, its dependence on full training features and $\mathcal{O}(n^3)$ matrix inversions limits its scalability to large datasets.

Rather than constructing each layer's operators directly from the features and compressing them into subspaces formed by the features themselves, CRATE compresses features into $K$ learnable subspaces $\boldsymbol{U}_{[K]}$ at each layer, enabling efficient batch-wise training on large datasets via backpropagation.

Specifically, a modified compression term $R_c(\boldsymbol{Z}|\boldsymbol{U}_{[K]})$ is proposed by incorporating $K$ trainable subspace matrices concatenated as $\boldsymbol{U}_{[K]} = [\boldsymbol{U}_1, \ldots, \boldsymbol{U}_K] \in \mathbb{R}^{d \times Kp}$. Each $\boldsymbol{U}_i \in \boldsymbol{R}^{d \times p}$ (with $p < d$) spans the basis of the $i$-th low-dimensional subspace. The modified compression term is defined as follows:

$$R_c(\boldsymbol{Z}|\boldsymbol{U}_{[K]}) = \frac{1}{2} \sum_{k=1}^{K} \log \det(\boldsymbol{I} + \frac{p}{n\epsilon^2}(\boldsymbol{U}_k^T \boldsymbol{Z})^T (\boldsymbol{U}_k^T \boldsymbol{Z})) \tag{2}$$

Here, $R_c(\boldsymbol{Z}|\boldsymbol{U}_{[K]})$ measures the compactness of representations $\boldsymbol{U}_k^T \boldsymbol{Z}$ within each low-dimensional subspace. CRATE approximates the gradient descent step of $R_c(\boldsymbol{Z}|\boldsymbol{U}_{[K]})$ to avoid the expensive matrix inverse, deriving the MSSA module of each layer:

$$\mathrm{MSSA}(\boldsymbol{Z}|\boldsymbol{U}_{[K]}) = \sum_{k=1}^{K} \boldsymbol{U}_k \boldsymbol{U}_k^T \boldsymbol{Z} \mathcal{S}((\boldsymbol{U}_k^T \boldsymbol{Z})^T (\boldsymbol{U}_k^T \boldsymbol{Z})) \tag{3}$$

where $\mathcal{S}(\cdot)$ denotes the softmax operator. However, MSSA may deviate from its intended compression objective: instead of reducing the compression term, it can theoretically increase it (Hu et al., 2024). This is because MSSA is derived from a second-order gradient approximation of $R_c(\boldsymbol{Z} \mid \boldsymbol{U}_{[K]})$, where the first-order term is omitted. Moreover, MSSA is computationally inefficient, as it requires computing pairwise similarities $(\boldsymbol{U}_k^T \boldsymbol{Z})^T (\boldsymbol{U}_k^T \boldsymbol{Z})$ across all input tokens, resulting in quadratic complexity.

ToST further reduces computational complexity by introducing an upper-bound variant of Eq. (2), which relies only on the diagonal elements of the correlation matrix $(\boldsymbol{U}_k^T \boldsymbol{Z} \mathrm{Diag}(\boldsymbol{\pi}_k) \boldsymbol{Z}^T \boldsymbol{U}_k)$. By performing a gradient descent step on this upper-bound variant, ToST derives the TSSA module. However, restricting attention to the diagonal elements ignores inter-dimensional correlations, which limits the ability to capture fine-grained structural information for feature updates (See Appendix B.1 for detailed analysis).

Moreover, neither model includes an explicit expansion module: MSSA is followed by a component primarily promoting feature sparsity, while TSSA is followed by a simple multi-layer perceptron (MLP) layer. Also, They undergo significant simplifications. Specifically, in both MSSA and TSSA, the part of the operator $[\boldsymbol{U}_1, \ldots, \boldsymbol{U}_K]$ that aggregates information across all class-specific subspaces is replaced by a single shared trainable matrix $\boldsymbol{W}$. Overall, these choices deviate from the principled design these models aimed at and may compromise their performance.

**Relation to other white-box models.** Recent works, such as contract- and-broadcast self-attention (CBSA) (Wen et al., 2026) and decoder for principled semantic segmentation (DEPICT) (Wen & Li, 2024), extend the exploration of

white-box attention architectures. In particular, CBSA provides a unified framework for several attention mechanisms, including softmax attention (Yu et al., 2023), linear attention (Katharopoulos et al., 2020), and channel attention (Hu et al., 2018; Woo et al., 2018; Wu et al., 2025). While both PACEAttention and CBSA reduce computational complexity, they differ fundamentally: PACEAttention exploits low-dimensional structure, whereas CBSA selects representative tokens. Furthermore, models including CBSA, DEPICT, and MSSA can be viewed under a similar softmax-based formulation, as shown in Eq. (3). This unified perspective suggests that the approximation issue discussed above may extend beyond MSSA to a broader family of white-box attention methods (Hu et al., 2024), which remains an open direction for further investigation.

## 3. PACEAttention

Our goal remains to expand all features ($R(\boldsymbol{Z})$ in Eq. (1)) and compress the representations within each subspace ($R_c(\boldsymbol{Z}|\boldsymbol{U}_{[K]})$ in Eq. (2)). Unlike prior operators that directly use or approximate the gradient of the objective function, we exploit the geometric interpretation of the gradient of $\log \det(\boldsymbol{I} + \boldsymbol{Z}\boldsymbol{Z}^T)$, a term included in both $R(\boldsymbol{Z})$ and $R_c(\boldsymbol{Z}|\boldsymbol{U}_{[K]})$, to redesign the expansion and compression modules. Consider a PSD matrix $\boldsymbol{M} = \boldsymbol{Z}\boldsymbol{Z}^T \in \mathrm{PSD}(d)$, $\log \det(\boldsymbol{I} + \boldsymbol{M}) = \sum_{i=1}^{d} \log(1 + \lambda_i(\boldsymbol{M}))$ where $\lambda_i(\boldsymbol{M})$ is the $i$-th largest eigenvalue of $\boldsymbol{M}$. To increase the value of $\log \det(\cdot)$, we can either increase the zero eigenvalues (corresponding to the null space $\mathrm{null}(\boldsymbol{Z}^T)$) or enhance the non-zero eigenvalues (corresponding to the column space $\mathrm{range}(\boldsymbol{Z})$). Hence we arrive at two core algorithmic ideas:

- For **feature expansion**, enlarge the eigenvalues associated with the null space, hence increasing the column space dimensionality and promoting separability.
- **Effective feature compression** can be realized by driving certain non-zero eigenvalues toward zero to reduce dimensionality of the column space.

### 3.1. Overview of PACEAttention

As illustrated in Figure 1(b), adding or subtracting the null space projection of the feature vector $\boldsymbol{z}$ moves $\boldsymbol{z}$ toward the null space or column space, thereby realizing feature expansion or compression. Let $\boldsymbol{Q}$ be the orthonormal basis of the identified column space $\mathrm{range}(\boldsymbol{Z})$. The projection of $\boldsymbol{z}$ onto the column space is $\boldsymbol{Q}\boldsymbol{Q}^T \boldsymbol{z}$, whereas the projection onto the null space is $(\boldsymbol{I} - \boldsymbol{Q}\boldsymbol{Q}^T)\boldsymbol{z}$ (Fessler & Nadakuditi, 2024). From this geometric insight, we propose the **PACEAttention** mechanism to achieve effective feature compression

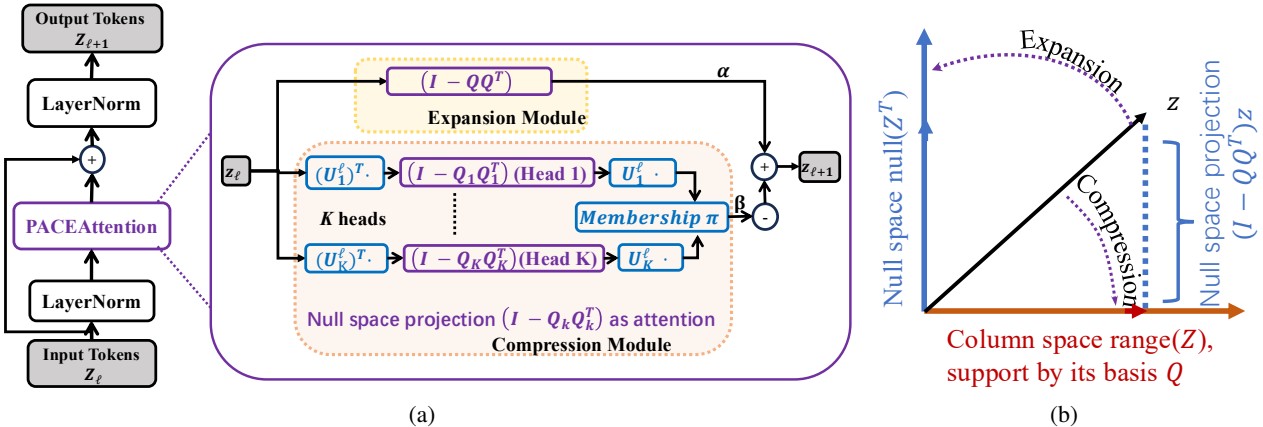

*Figure 1.* (a) One layer of PACENet. (b) Illustration of expansion and compression.

and expansion at each layer:

$$\boldsymbol{z}_{\ell+1} = \boldsymbol{z}_\ell + \underbrace{\alpha(\boldsymbol{I} - \boldsymbol{Q}\boldsymbol{Q}^T)\boldsymbol{z}_\ell}_{\text{Expansion}}$$

$$\underbrace{-\beta \sum_{k=1}^{K} \pi_k \boldsymbol{U}_k (\boldsymbol{I} - \boldsymbol{Q}_k \boldsymbol{Q}_k^T) \boldsymbol{U}_k^T \boldsymbol{z}_\ell}_{\text{Compression}}. \quad (4)$$

Here, $\boldsymbol{Q}_k$ denotes the orthonormal basis of $\text{range}(\boldsymbol{U}_k^T \boldsymbol{Z})$. Two learnable parameters, $\alpha$ and $\beta$, provide a self-adaptive mechanism that determines the relative contribution of expansion and compression in each layer. For brevity of presentation, we omit the layer index $\ell$ on all layer-specific parameter matrices ($\boldsymbol{Q}_{(\ell)}, \boldsymbol{U}_{(\ell)}$, etc.) unless explicitly required. We refer to the PACEattention-based network model as "PACENet". Figure 1(a) gives an overview of a layer of PACENet. The resulting architecture remains remarkably simple. Each feature is transformed by PACEAttention and subsequently added back to the input via a residual connection. The expansion module aims to enlarge the coding rate of all features $R(\boldsymbol{Z})$ in Eq. (1), while the compression module seeks to decrease the coding rate of intra-group features $R_c(\boldsymbol{Z}|\boldsymbol{U}_{[K]})$ in Eq. (2).

Unlike the expansion module that directly updates features via null space projection, trainable parameter matrices $\boldsymbol{U}_{[K]}$ in the compression module can be viewed as $K$ bases. Each feature $\boldsymbol{z}$ is first projected onto the basis $\boldsymbol{U}_k$ by multiplying by $\boldsymbol{U}_k^T$, then updated according to the null space projection $(\boldsymbol{I} - \boldsymbol{Q}_k \boldsymbol{Q}_k^T)$, and finally projected back to the standard basis by multiplying by $\boldsymbol{U}_k$. The membership $\boldsymbol{\pi} = [\pi_1, \ldots, \pi_K]$, computed from the similarity between $\boldsymbol{z}$ and bases $\boldsymbol{U}_{[K]}$, gives probabilities of the token belonging to each group, guiding its update toward the corresponding group.

### 3.2. Construct Expansion & Compression Modules based on Null Space Projection

The expansion module is designed to increase the coding rate $R(\boldsymbol{Z})$ layer by layer. This is achieved in two steps: (1) project each feature $\boldsymbol{z} \in \boldsymbol{Z}$ onto the null space $\text{null}(\boldsymbol{Z}^T)$, and (2) add the projection to each feature so that it moves towards the null space of the overall features. Hence, after obtaining the orthonormal basis $\boldsymbol{Q} \in \mathbb{R}^{d \times Kr}$ of $\text{range}(\boldsymbol{Z})$, the expansion update can be denoted as:

$$\boldsymbol{z}_{\ell+1} = \boldsymbol{z}_\ell + (\boldsymbol{I} - \boldsymbol{Q}\boldsymbol{Q}^T)\boldsymbol{z}_\ell \in \mathbb{R}^d. \quad (5)$$

Here $K$ is the number of heads and also the number of subspaces, $r$ is the the rank of each subspace; $Kr$ gives the dimension of the overall feature column space, $Kr \leq d$.

When it comes to the compression module, the situation is different as the object to be compressed in Eq. (2) is the *code* (i.e., a low-dimensional representation) of the token feature $\boldsymbol{\alpha}_k = \boldsymbol{U}_k^T \boldsymbol{z}$ rather than the feature $\boldsymbol{z}$ (see Appendix B.2 for a formal description). The *codes* of all features on subspace $\boldsymbol{U}_k$ can be denoted as $\boldsymbol{A}_k = \boldsymbol{U}_k^T \boldsymbol{Z} \in \mathbb{R}^{p \times n}$ and its column space is $\boldsymbol{Q}_k \in \mathbb{R}^{p \times r}$. Hence, for a *code* $\boldsymbol{\alpha}_k$ of token feature, the projection onto the null space $\text{null}(\boldsymbol{A}_k^T)$ and column space $\text{range}(\boldsymbol{A}_k)$ are $(\boldsymbol{I} - \boldsymbol{Q}_k \boldsymbol{Q}_k^T)\boldsymbol{\alpha}_k$ and $\boldsymbol{Q}_k \boldsymbol{Q}_k^T \boldsymbol{\alpha}_k$, respectively. Hence, by subtracting the null space projection, we can achieve the goal of compressing *code* $\boldsymbol{\alpha}_k$ toward their corresponding subspace.

$$\boldsymbol{\alpha}_k^{\ell+1} = \boldsymbol{\alpha}_k^\ell - (\boldsymbol{I} - \boldsymbol{Q}_k \boldsymbol{Q}_k^T)\boldsymbol{\alpha}_k^\ell \in \mathbb{R}^p \quad (6)$$

The projection onto the column spaces $\{\boldsymbol{Q}_k \boldsymbol{Q}_k^T \boldsymbol{\alpha}_k^\ell\}_{k=1}^K$ can be converted to a distribution of membership $\boldsymbol{\pi}$ as follows:

$$\boldsymbol{\pi} = \text{softmax} \left( \frac{1}{2\eta} \begin{bmatrix} ||\boldsymbol{Q}_1 \boldsymbol{Q}_1^T \boldsymbol{\alpha}_1^\ell||^2 \\ \vdots \\ ||\boldsymbol{Q}_K \boldsymbol{Q}_K^T \boldsymbol{\alpha}_K^\ell||^2 \end{bmatrix} \right) \quad (7)$$

$$= [\pi_1, \ldots, \pi_K] \in \mathbb{R}^K$$

This quantifies the probability that a token belongs to each subspace, which can be used as weights to guide each feature to update towards its corresponding subspace. Finally, we obtain the following update formula for feature compression (See Appendix B.2 for details):

$$\boldsymbol{z}_{\ell+1} \approx \boldsymbol{z}_\ell - \sum_{k=1}^K \pi_k \boldsymbol{U}_k (\boldsymbol{I} - \boldsymbol{Q}_k \boldsymbol{Q}_k^T) \boldsymbol{U}_k^T \boldsymbol{z}_\ell \qquad (8)$$

### 3.3. Obtain the Basis of Column Space Efficiently

One key operation in PACEAttention is to identify the column space of the feature matrix, which is typically considered computationally intensive. In this work, we choose to leverage a randomization method to capture column space efficiently (Halko et al., 2011). Once the basis $\boldsymbol{Q}$ of the column space is obtained, the projection onto the null space is given by $(\boldsymbol{I} - \boldsymbol{Q}\boldsymbol{Q}^T)$. Specifically, suppose that we seek the column space of feature matrix $\boldsymbol{Z} = [\boldsymbol{z}_1 \cdots \boldsymbol{z}_n] \in \mathbb{R}^{d \times n}$ with a specified rank $r$. A random matrix $\boldsymbol{\Omega} = [\boldsymbol{\omega}_1 \cdots \boldsymbol{\omega}_r] \in \mathbb{R}^{n \times r}$ can be multiplied with $\boldsymbol{Z}$: $\boldsymbol{Y} = [\boldsymbol{Z}\boldsymbol{\omega}_1 \cdots \boldsymbol{Z}\boldsymbol{\omega}_r] = [\boldsymbol{y}_1 \cdots \boldsymbol{y}_r] \in \mathbb{R}^{d \times r}$. Due to the randomness, the random vectors $[\boldsymbol{\omega}_1 \cdots \boldsymbol{\omega}_r]$ form a linearly independent set with high probability. The resulting linear combinations $[\boldsymbol{y}_1 \cdots \boldsymbol{y}_r]$ lie in the column space of $\boldsymbol{Z}$ and are also linearly independent, thereby spanning the column space of $\boldsymbol{Z}$. This also plays a role in dimensionality reduction, thereby resulting in a reduced-dimension compression space. Subsequently, given the higher efficiency of Cholesky decomposition compared to QR decomposition, we employ Cholesky decomposition to further process the Gram matrix $\boldsymbol{Y}^T\boldsymbol{Y}$ to obtain an orthonormal basis $\boldsymbol{Q}$ (see Appendix B.3 for details).

**Complexity analysis.** Since our expansion module is aligned with the MLP module in ToST, it is appropriate to compare only the proposed compression module with the TSSA module. The time complexity of the compression module is almost equivalent to that of the TSSA module ($\mathcal{O}(pn)$), which is a *linear-time* attention, except that our module includes an additional cost for Cholesky decomposition ($\mathcal{O}(\frac{1}{3}r^3)$) that aims to capture informative structure for feature updates. Since the rank $r$ of random matrix $\boldsymbol{\Omega}$ is predefined as constant, the time cost of the Cholesky decomposition remains stable. Hence, as Figure 2(a) shows, although PACENet has slightly higher time complexity when the number of tokens is small. As the number of tokens increases, the time complexity of PACENet becomes lower than that of ToST. After removing the Cholesky decomposition step (PACENet w/o Cholesky), our model shows lower time complexity, which confirms that the source of additional time cost comes from the Cholesky decomposition. In contrast, the attention operators of CRATE exhibit quadratic time complexity of $\mathcal{O}(pn^2)$. Besides, Figure 2(b)

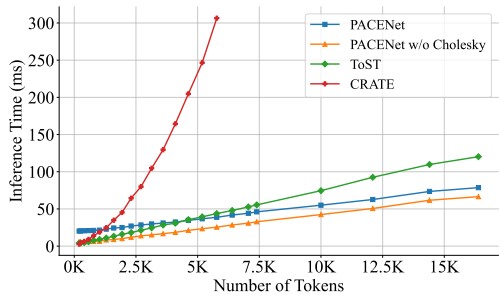

(a)

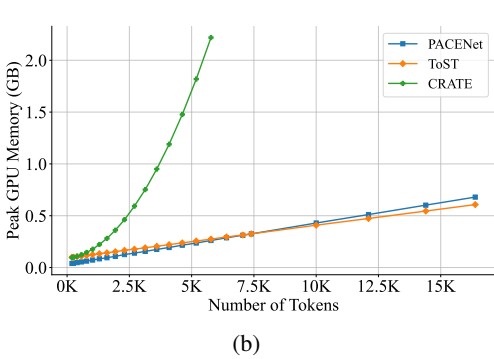

(b)

*Figure 2.* Comparison on (a) inference time, and (b) memory usage when the number of input tokens increases. All models are measured with 12 layers, $d = 384$, and $K = 8$.

suggests that our model exhibits linear memory usage as the number of tokens $n$ increases. Detailed analysis is provided in Appendix B.4.

## 4. Experiments

We conduct experiments on both toy data and real-world datasets to verify and study the properties and performance of PACENet. We adopt a straightforward implementation that strictly follows our formulation. Therefore, demonstrating that the current implementation of PACENet outperforms existing highly engineered architectures is not the goal of this work. Rather, our empirical studies aim to provide answers and evidence for the following questions:

1. Are the expansion and compression modules effective?

2. Does compression via null-space subtraction give any advantage over direct column-space projection?

3. Does the PACENet provide better interpretability?

4. Does the PACENet exhibit self-adaptivity?

Our framework uses PACEAttention as the backbone, while other components, such as tokenization and positional encoding, largely follow the ToST implementation. To address the above questions, we perform experiments across models of varying scales by adjusting the number of attention

heads $K$, the token dimension $d$, and the number of layers $L$. Due to space limitations, the details on practical implementation and configurations are provided in Appendices C.1 and C.2. Additional ablation studies and the pseudocode of PACEAttention are provided in Appendices D and E.

**Datasets and training configuration.** We pre-train the PACENet models on ImageNet-1K dataset (Deng et al., 2009; Beyer et al., 2020). The pre-trained models are subsequently adapted through fine-tuning on downstream tasks using CIFAR-10/100 (Krizhevsky et al., 2009), Oxford Flowers (Nilsback & Zisserman, 2008), and Oxford-IIT-Pets (Parkhi et al., 2012) datasets. We set the initial values of learnable parameters to $\alpha = 0.1$ and $\beta = 0.1$ for each layer, with $r = 20$ and 500 epochs for pre-training. More training details are provided in Appendix C.3.

### 4.1. Verification on Toy Data

**Effective expansion and compression.** We begin by using synthetic toy data to validate the effectiveness of compression and expansion modules. Specifically, we assume that each feature $z$ in $Z \in \mathbb{R}^{d \times n}$ is generated from one of $K = 6$ mutually orthogonal subspaces $U_{[K]}$, where each subspace is supported by an orthonormal basis $U_k \in \mathbb{R}^{d \times p}$.

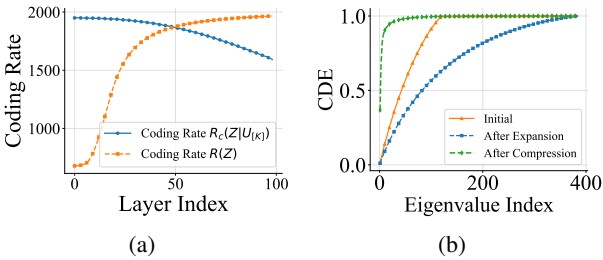

*Figure 3.* (a) Layer-wise changes in the coding rate; (b) Changes in the cumulative distribution of eigenvalues (CDE) of $ZZ^T$.

We set $d = 384$ and $p = 64$, and construct each sample as $z = U_k \alpha$ by activating only the first 20 dimensions of $U_k$ through $\alpha$. In total, we generate $n = 1000$ samples. The resulting dataset has an overall rank of 120, with each subspace being 20-dimensional.

We apply the compression and expansion operations to this data separately. The layer-wise coding rates are computed according to the expansion term $R(Z)$ in Eq. (1) and the modified compression term $R_c(Z|U_{[K]})$ in Eq. (2). As shown in Figure 3(a), the proposed expansion and compression modules successfully achieve their intended design objectives, with $R(Z)$ being incrementally expanded and $R_c(Z|U_{[K]})$ being gradually reduced across layers. Figure 3(b) shows the cumulative distribution of eigenvalues,

which reflects how many dimensions carry informative content out of the 384 dimensions. Initially, the dataset has a rank of 120, meaning that the first 120 dimensions capture all the energy (i.e., correspond to nonzero eigenvalues). As shown in Figure 3(b), the proposed modules achieve effective expansion and compression: expansion is realized by increasing the eigenvalues associated with the null space, thereby enlarging the effective dimensions, while compression narrows the dimensions of the column space.

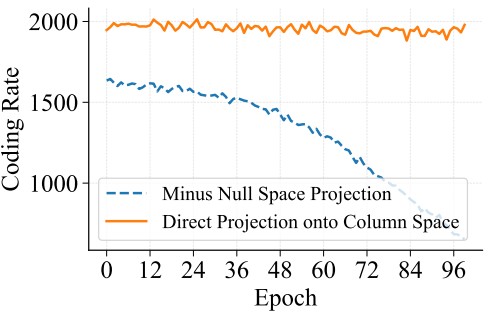

*Figure 4.* Compression by subtracting null-space projection vs. via column-space projection.

**More effective compression by subtracting null space projection.** We choose to subtract the null space projection (i.e., $I - Q_k Q_k^T$) rather than directly projecting onto the column space (i.e., $Q_k Q_k^T$) for feature compression at each layer. As shown in Figure 4, compression by subtracting null space projection enables effective multi-epoch optimization. The reason is that compression via subtracting the null-space projection yields a skip-connection–like update, i.e., $z_{\ell+1} = z_\ell - $ (null-space projection), which is more favorable for multi-epoch and layer-wise optimization. In contrast, directly projecting onto the column space results in a structure like $z_{\ell+1} = $ (column-space projection). This structure blocks the input information of each layer from flowing to subsequent layers, which undermines both layer-wise optimization and multi-epoch training (See Appendix D.5 for more details).

### 4.2. Experiments on Real-World Visual Data

**Training & fine-tuning results.** As approximately two-thirds of the parameters in ToST belong to the MLP layers, which is not a principled module derived from the MCR$^2$, we evaluate ToST *with* and *without* MLP separately to enable a fair comparison. The results of CRATE are directly sourced from the original papers, as this method computationally prohibitive to reproduce in our setting.

As shown in Figure 5, PACENet consistently outperforms CRATE across comparable parameter scales. For ToST, we observe that ToST w/o MLP, which relies solely on the principled TSSA module, does not scale well with increasing

model size, with training quickly terminating due to numerical errors. Consequently, we are only able to obtain results at a small parameter scale.

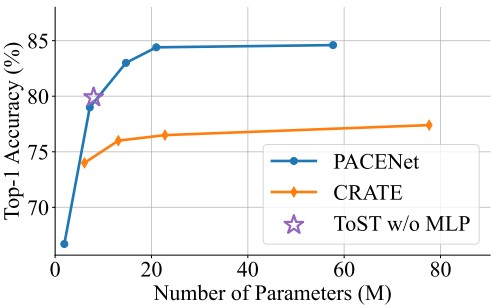

*Figure 5.* Top-1 accuracy of three principled models on ImageNet-Real across parameter scales.

Table 1 presents the top-1 accuracy of PACENet on the ImageNet-1K dataset and fine-tuning accuracy across several smaller datasets. For CIFAR-10/100, we report fine-tuning results over 200 epochs, while for Oxford Flowers-102 and Oxford-IIIT-Pets, we report results over 30 epochs. Overall, PACENet achieves performance comparable to ToST w/ MLP. Specifically, when comparing PACENet-S with ToST-T, PACENet-S delivers better performance, and we further observe that PACENet-S converges faster during fine-tuning on Oxford Flowers-102. In addition, when comparing PACENet-M with ToST-S, PACENet-M attains comparable results while using only two-thirds of the parameters. PACENet-B+ does not achieve accuracy gains comparable to those observed in ToST w/ MLP under increased parameter budgets. We note that the scaling advantage of ToST w/ MLP largely stems from its MLP component rather than the principled TSSA module itself. As shown in Figure 5, ToST w/o MLP struggles to scale beyond small model sizes. We provide an empirical analysis of MLP capacity allocation in ToST at the end of Section 4.3.

**Improved interpretability in membership visualization.** At each layer, Eq. (7) estimates the soft membership vector $\pi$ of each token, representing the probability that the token is assigned to the $k$-th group (or subspace). Hence, we obtain $K$ clusters over the $N$ tokens. As Figure 6 shows, following the design principle of compressing tokens into different subspaces, the PACENet automatically learns object segmentation without complex self-supervised learning recipes or segmentation-related annotations. In particular, heads 3 and 4 in PACENet-B not only identify objects, but also enable more **fine-grained segmentation**. For example, in the building image (demo 5), head 3 captures the overall framework, while head 4 focuses on the detailed structure of individual floors. Similarly, in demo 4, head 2 identifies the tree while head 3 attends to the door. In contrast, ToST-S heads mainly focus on coarse object–background

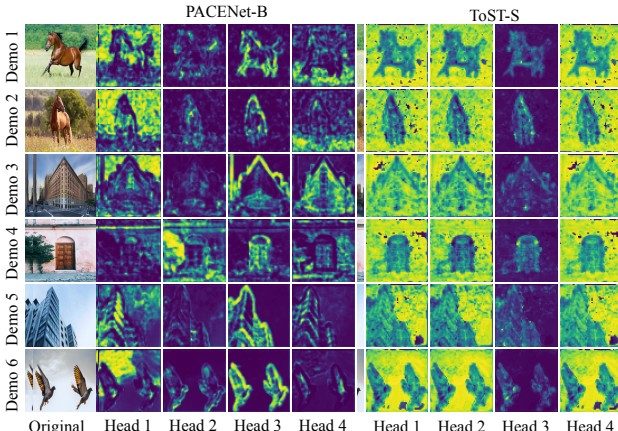

*Figure 6.* Membership distribution $\pi$ of PACENet-B (*left*) and ToST-S (*right*). Different heads in PACENet-B attend to distinct object parts, enabling more fine-grained structural modeling. We visualize the head-wise membership $\pi_k$, reshaped to $\sqrt{N} \times \sqrt{N}$ for $N$ tokens. Results are shown for layer 3 in PACENet-B and layer 9 in ToST-S, which provide the clearest visualizations. Similar trends are observed in other layers.

separation. Note that bright yellow indicates the attended regions. Only one attention head (head 3) in ToST focuses on the target object, while the remaining heads attend almost uniformly to background regions, exhibiting little diversity. This suggests that multiple heads do not effectively capture complementary or discriminative information.

### 4.3. Hyperparameter Analysis

We introduce two learnable weight parameters, $\alpha$ and $\beta$, for the expansion module and compression module, respectively, in Eq. (4), and one predefined hyperparameter $r$ for the random matrix $\Omega$. As mentioned in Section 3.3, $r$ controls the dimensionality of the extracted features, thereby determining the volume of the target subspace for feature compression. To investigate the effect of the hyperparameter $r$, we evaluate our approach on the CIFAR-10 dataset with two model variants of different capacities: PACENet-S (7.2M) and PACENet-B (21M).

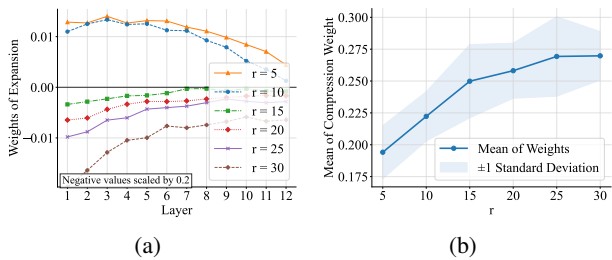

*Figure 7.* (*Left*) Expansion weights $\alpha$ across layers for different $r$ values. (*Right*) Mean of compression weights $\beta$ across layers for different $r$ values. Results are obtained from PACENet-S trained on the CIFAR-10 dataset for 400 epochs.

*Table 1.* Top 1 accuracy of PACENet across datasets with different model sizes. For ImageNet and ImageNet ReaL, we directly evaluate the top-1 accuracy. For other datasets, we pre-train the model on ImageNet and fine-tune it. Here, we report the results of ToST with MLP across different model scales, as ToST without MLP cannot be evaluated at larger parameter scales due to scalability limitations.

| Datasets | PACENet-S | PACENet-M | PACENet-B+ | ToST-T | ToST-S | ToST-M | CRATE-B | CRATE-L |
|---|---|---|---|---|---|---|---|---|
| # parameters | 7.2M | 14.7M | 57.7M | 5.8M | 22.6M | 68.1M | 22.8M | 77.6M |
| ImageNet | 67.9 | 73.1 | 75.6 | 64.9 | 77.5 | 79.6 | 70.8 | 71.3 |
| ImageNet ReaL | 79.0 | 83.0 | 84.6 | 76.8 | 86.1 | 87.0 | 76.5 | 77.4 |
| CIFAR10 | 94.1 | 95.2 | 96.2 | 94.7 | 97.4 | 98.0 | 96.8 | 97.2 |
| CIFAR100 | 77.3 | 81.2 | 82.2 | 77.2 | 85.4 | 86.1 | 82.7 | 83.6 |
| Oxford Flowers-102 | 75.5 | 83.4 | 99.5 | 49.1 | 95.4 | 98.8 | 88.7 | 88.3 |
| Oxford-IIIT-Pets | 92.6 | 95.7 | 99.2 | 86.0 | 98.6 | 99.5 | 85.3 | 87.4 |

**Self-adaptive capabilities of expansion and compression.** We first analyze the learned expansion–compression weights under different values of $r$. As shown in Figure 7(a), when $r \leq 10$, the model adaptively assigns larger expansion weights to early layers and smaller weights to deeper layers. This self-adaptive strategy promotes inter-group feature separation in early layers while facilitating intra-group compression in later layers, consistent with prior findings(Yu et al., 2024). For the compression module, as shown in Figure 7(b), the mean weight increases as the predefined compression space becomes larger, suggesting that the network progressively favors feature compression.

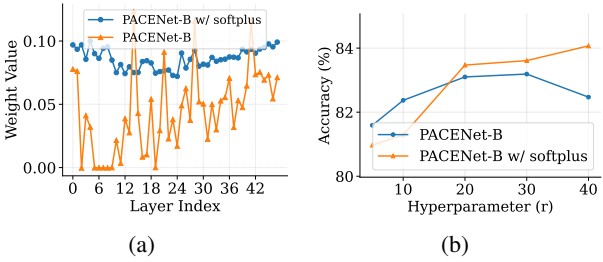

(a)                                              (b)

*Figure 8.* (a) The weight distribution of compression modules. We train the PACENet-B on CIFAR-10. (b) Softplus trick stabilizes the effect of tuning the hyperparameter $r$.

However, as shown in Figure 7(a), for larger values of $r$, expansion weights can become negative, which leads the expansion operator to function as a form of compression. In addition, as shown in Figure 8(a), the compression module weights in certain layers are zero in larger model like PACENet-B. Together, these observations suggest that although the dynamics are flexible, unconstrained weights can result in behaviors that are not fully aligned with the design intent, potentially leaving some layers inactive.

To preserve self-adaptive behavior while enforcing consistency with the design intent of expansion and compression, we impose a non-negativity constraint on the expansion weights using a softplus function (Dugas et al., 2000). As shown in Figure 8(a), the softplus trick ensures that all lay-

ers remain active and reduces the fluctuation of the weights across layers. In addition, as shown in Figure 8(b), this trick makes the effect of tuning the hyperparameter $r$ more stable: a larger value of $r$ results in higher accuracy.

**Choice of r.** We empirically set $r = 20$ based on a trade-off between computational cost and model accuracy. A basic principle is that the dimension of the overall feature column space, $Kr$, should not exceed the feature dimension $d$. Consequently, $r$ must satisfy $r \leq p = d/K$, where $p$ is the dimension of each head. Furthermore, Pope et al.(2021) estimated the intrinsic dimensionality of the ImageNet dataset and showed that its intrinsic dimension does not exceed 43. In addition, Figure 8(b) demonstrates that increasing $r$ beyond 20 yields limited additional gains. Finally, although we uniformly set $r = 20$ across all layers in the current model, singular value spectrum analysis shows that the top-20 components capture the vast majority of feature energy (from 89.5% in initial layers to over 99% in deeper layers). These observations provide empirical support for choosing $r = 20$ as a favorable balance between representation capacity and computational complexity.

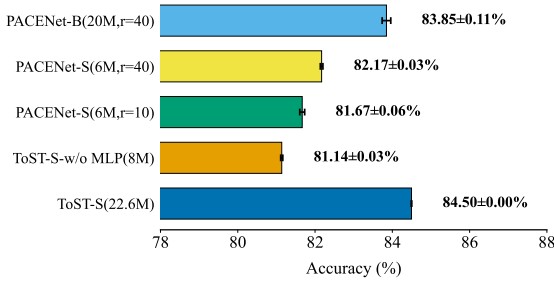

*Figure 9.* A comparison with ToST on CIFAR-10

**Empirical analysis of capacity allocation.** We empirically analyze capacity allocation in ToST by removing its MLP components, confirming that approximately two-thirds of the parameters originate from the MLPs, as shown in Figure 9 through the comparison between ToST-S-w/o MLP

(8M) and ToST-S (22.6M). Besides, ToST-S-w/o MLP exhibits the lowest accuracy of 81.14%. In contrast, PACENet-S, with an even smaller parameter size (6M), surpasses ToST-S-w/o MLP. The figure further shows that increasing $r$ or the model size leads to corresponding improvements in accuracy. Similar trends are also observed on the CIFAR-100 dataset (See Appendix D.2).

## 5. Conclusion

In this work, we propose a principled attention mechanism called PACEAttention that updates features by operating in low-dimensional subspace, avoiding both the high computational complexity and the overly simplified operations adopted in prior principled approaches. This design is motivated by geometric insights from the gradient of the MCR$^2$ objective, which naturally give rise to two explicit modules for feature expansion and compression. In particular, both modules leverage null-space projections as the core information for feature updates, exhibiting clear geometric behavior. As a result, the proposed PACENet offers improved interpretability and enables more fine-grained representation modeling. Additionally, PACENet's self-adaptivity enables a macro-level understanding of how the model coordinates expansion and compression across layers. Compared with a "white-box" variant of ToST (with non-principled MLP layers removed) and CRATE (whose principled module is less aligned with the intended design), our framework adheres more closely to the white-box design principle and achieves better performance.

**Limitations and future works.** Our evaluation focuses on image classification benchmarks, and the hyperparameter controlling low-dimensional structure extraction still requires manual tuning. Future directions include extending PACEAttention to other tasks and modalities, particularly investigating how the learned representations perform on dense prediction tasks such as segmentation, as well as exploring principled approaches to automatic capacity control.

## Acknowledgements

This research was partially supported by Key Research Project of Quancheng Laboratory, China under Grant No. QCL20250103, Shandong Provincial Key Projects of Basic Research, China under Grant No.ZR2022ZD01, TaiShan Scholars Program under Grants No.tstp20240828, China Scholarship Council under Grant No.202208370034.

The author(s) wish to acknowledge use of the eResearch Infrastructure Platform hosted by the Crown company, Research and Education Advanced Network New Zealand (RE-ANNZ) Ltd., and funded by the Ministry of Business, Innovation & Employment. https://www.reannz.co.nz

## Impact Statement

This paper presents work aimed to advance the theory of Machine Learning. We propose a principled attention mechanism from an information-theory based objective function. Except for academic values, we do not anticipate any direct social or ethical implications.

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

## A. Summary of Feature Update Equations in the ReduNet Family

To ensure that the paper is self-contained, we summarize the core feature update equations of the ReduNet family in this section. ReduNet constructs its layer by directly following the gradient of MCR$^2$ objective in Eq. (1), denoted as:

$$\left.\frac{\partial \Delta R}{\partial \boldsymbol{Z}}\right|_{\boldsymbol{Z}_\ell} = \underbrace{\boldsymbol{E}_\ell}_{\text{Expansion}} \boldsymbol{Z}_\ell - \sum_{k=1}^{K} \gamma_k \underbrace{\boldsymbol{C}_\ell^k}_{\text{Compression}} \boldsymbol{Z}_\ell \text{Diag}(\boldsymbol{\pi}_k) \tag{9}$$

Here, the expansion operator $\boldsymbol{E}_\ell = \alpha(\boldsymbol{I} + \alpha\boldsymbol{Z}_\ell\boldsymbol{Z}_\ell^T)^{-1} \in \mathbb{R}^{n \times n}$ and the compression operator $\boldsymbol{C}_\ell^k = \alpha_k(\boldsymbol{I} + \alpha_k\boldsymbol{Z}_\ell\text{Diag}(\boldsymbol{\pi}_k)\boldsymbol{Z}_\ell^T)^{-1} \in \mathbb{R}^{n \times n}$, where $\alpha = \frac{d}{n\epsilon^2}$, $\alpha_k = \frac{d}{n_k\epsilon^2}$, $\gamma_k = \frac{n_k}{n}$. Hence, the parameters of the $\ell$-th layer (that is, $\boldsymbol{E}_\ell$ and $\{\boldsymbol{C}_\ell^k\}_{k=1}^K$) are constructed using the updated features $\boldsymbol{Z}_\ell$ from the previous layer. Thereby, for the $i$-th feature $\boldsymbol{z}_\ell$, the update equation is :

$$\boldsymbol{z}_{\ell+1} = \boldsymbol{z}_\ell + \eta\left(\boldsymbol{E}_\ell\boldsymbol{z}_\ell - \sum_{k=1}^{K} \gamma_k \boldsymbol{C}_\ell^k \boldsymbol{z}_\ell^i \boldsymbol{\pi}_k(i)\right) \tag{10}$$

Here, $\eta$ is the learning rate.

MSSA in CRATE (Yu et al., 2023) is derived from the approximation of the gradient of Eq. (2). Each layer of CRATE consists of two optimization steps: the MSSA step and the ISTA-like sparsification step (Beck & Teboulle, 2009). Specifically, the feature update at each layer can be formulated as follows:

$$\boldsymbol{Z}_{\ell+\frac{1}{2}} = \boldsymbol{Z}_\ell - \eta\nabla R_c(\boldsymbol{Z}_\ell|\boldsymbol{U}_\ell) \approx \boldsymbol{Z}_\ell + \eta\text{MSSA}(\boldsymbol{Z}_\ell|\boldsymbol{U}_\ell) \tag{11}$$

$$\boldsymbol{Z}_{\ell+1} = \text{ISTA}(\boldsymbol{Z}_{\ell+\frac{1}{2}}) \tag{12}$$

ToST (Wu et al., 2025) seeks to minimize the upper bound of the compression term. For a matrix $\boldsymbol{M} \in \mathbb{R}^{m \times n}$, we denote by $\boldsymbol{M}^{\odot 2} \in \mathbb{R}^{m \times n}$ the element-wise square of $\boldsymbol{M}$. For a function $f : \mathbb{R} \to \mathbb{R}$, let $f[\boldsymbol{v}] \in \mathbb{R}^n$ be the element-wise application of $f$ to the entries of $\boldsymbol{v}$. The variational objective is:

$$R_{c,f}^{\text{var}}(\boldsymbol{Z}, \boldsymbol{\Pi}|\boldsymbol{U}) = \frac{1}{2}\sum_{k=1}^{K} \frac{n_k}{n} \sum_{i=1}^{d} f\left(\frac{1}{n_k}(\boldsymbol{U}_k^T\boldsymbol{Z}\text{Diag}(\boldsymbol{\pi_k})\boldsymbol{Z}^T\boldsymbol{U}_k)_{ii}\right) \tag{13}$$

Here, $f(x) = \log(1 + (d/\epsilon^2)x)$. Therefore, a linear-time attention TSSA is obtained from the gradient of Eq. (13):

$$\nabla_{\boldsymbol{Z}}R_{c,f}^{\text{var}} = \frac{1}{n}\sum_{k=1}^{K} \boldsymbol{U}_k \underbrace{\text{Diag}\left(\nabla f\left[(\boldsymbol{U}_k^T\boldsymbol{Z})^{\odot 2}\frac{\boldsymbol{\pi}_k}{\langle\boldsymbol{\pi}_k, \boldsymbol{1}\rangle}\right]\right)}_{\text{D}(\boldsymbol{Z},\boldsymbol{\pi}_k|\boldsymbol{U}_k)} \boldsymbol{U}_k^T\boldsymbol{Z}\text{Diag}(\boldsymbol{\pi}_k) \tag{14}$$

$$\boldsymbol{z}_{\ell+1} = \boldsymbol{z}_\ell - \frac{\eta}{n}\sum_{k=1}^{K} \boldsymbol{\pi}_k(i)\boldsymbol{U}_k\text{D}(\boldsymbol{Z}, \boldsymbol{\pi}_k|\boldsymbol{U}_k)\boldsymbol{U}_k^T\boldsymbol{z}_\ell \tag{15}$$

The $\nabla f[\cdot]$ is applied to each element of the vector in the bracket. Each layer of ToST still follows a two-step optimization design similar to CRATE, with TSSA followed by an MLP layer.

## B. Theoretical Analysis

### B.1. Analysis of TSSA

TSSA uses second-moment statistics for feature update as shown in Eq. (16).

$$\text{D}(\boldsymbol{Z}, \boldsymbol{\pi}_k|\boldsymbol{U}_k) = \text{Diag}\left(\nabla f\left[(\boldsymbol{U}_k^T\boldsymbol{Z})^{\odot 2}\frac{\boldsymbol{\pi}_k}{\langle\boldsymbol{\pi}_k, \boldsymbol{1}\rangle}\right]\right) \tag{16}$$

Here, $f(x) = \log(1 + x)$. The $\nabla f[\cdot]$ is applied to each element of the vector in the bracket. To facilitate understanding, we demonstrate TSSA's operation through a simple example involving two samples with two-dimensional features $\boldsymbol{Z} = [\boldsymbol{z}_1, \boldsymbol{z}_2] \in \mathbb{R}^{2 \times 2}$ and membership vector $\boldsymbol{\pi}_k = [\pi_1, \pi_2]^T \in \mathbb{R}^{2 \times 1}$ represents the probability that each sample belongs to $k$-th group.

$$\mathrm{Diag}(\boldsymbol{Z}^{\odot^2} \boldsymbol{\pi}_k) = \mathrm{Diag}\left(\begin{bmatrix} z_{11}^2 & z_{21}^2 \\ z_{12}^2 & z_{22}^2 \end{bmatrix} \begin{bmatrix} \pi_1 \\ \pi_2 \end{bmatrix}\right) = \mathrm{Diag}\left(\begin{bmatrix} z_{11}^2 \pi_1 + z_{21}^2 \pi_2 \\ z_{12}^2 \pi_1 + z_{22}^2 \pi_2 \end{bmatrix}\right) \tag{17}$$

This demonstrates that the $i$-th diagonal element of TSSA captures the sum of squared $i$-th dimensional components across samples in the $k$-th group, thereby providing statistical information that informs feature update mechanisms. Essentially, this is equivalent to the "weighted" diagonal elements of the covariance matrix $\boldsymbol{Z}\mathrm{Diag}(\boldsymbol{\pi}_k)\boldsymbol{Z}^T$, thus ignoring the correlations between different dimensions. This can be formalized as follows:

$$\boldsymbol{Z}\mathrm{Diag}(\boldsymbol{\pi}_k)\boldsymbol{Z}^T = \begin{bmatrix} z_{11}^2 \pi_1 + z_{21}^2 \pi_2 & z_{11}z_{12}\pi_1 + z_{21}z_{22}\pi_2 \\ z_{12}z_{11}\pi_1 + z_{22}z_{21}\pi_2 & z_{12}^2 \pi_1 + z_{22}^2 \pi_2 \end{bmatrix} \tag{18}$$

We also give a toy data example to show that such simple statistics are inadequate for capturing fine-grained structures

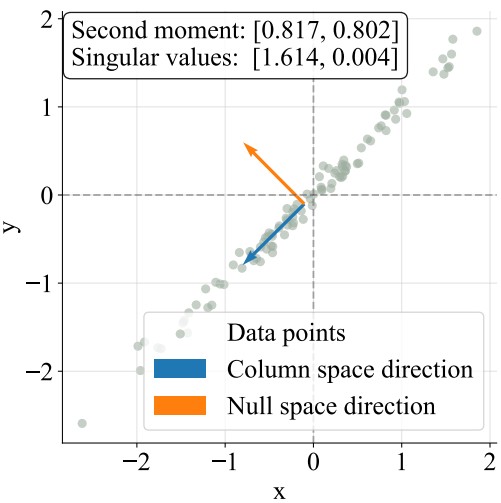

*Figure 10.* Second-moment statistic vs Singular values

that can effectively guide feature compression. Consider that 100 data points $(x, y)$ are generated from the linear model $y = x + \epsilon$, where $\epsilon \sim \mathcal{N}(0, 0.1)$ is a Gaussian noise. Since the linear dependence between variables $x$ and $y$, the rank of the generated data matrix should be 1. As Figure 10 shows, the first singular value obtained by singular value decomposition (SVD) is significantly larger than the second, indicating that the true dimensionality of the data is one-dimensional. In contrast, the second moment statistics do not reveal a clear distinction between the two dimensions of the data.

### B.2. Proof of the Compression Formula

In the main text, we noted that the compression module operates not directly on the token feature $\boldsymbol{z}$, but rather on its code $\boldsymbol{\alpha}_k = \boldsymbol{U}_k^T \boldsymbol{z}$ with respect to the basis $\boldsymbol{U}_k$. Here, we provide a formal mathematical description of the assumption behind this. Specifically, the definition of $R_c(\boldsymbol{Z} \mid \boldsymbol{U}_{[K]})$ is grounded in the **union of $K$ low-dimensional subspace models**, involving $K$ orthonormal basis matrices. Assume that $\boldsymbol{U}_{[K]} = [\boldsymbol{U}_1, \ldots, \boldsymbol{U}_K] \in \mathbb{R}^{d \times Kp}$ is a set of orthonormal basis matrices for $K$ subspaces with $\boldsymbol{U}_k \in \mathbb{R}^{d \times p}$. We say that a token feature $\boldsymbol{z} = \boldsymbol{U}_k \boldsymbol{\alpha} \in \mathbb{R}^d$ lies on the union of subspaces supported by $\boldsymbol{U}_{[K]}$. From an information theory perspective, $\boldsymbol{U}_{[K]}$ can be viewed as codebooks and the vectors $\boldsymbol{\alpha} = \boldsymbol{U}_k^T \boldsymbol{z} \in \mathbb{R}^p$ can be the codes of the token feature $\boldsymbol{z}$ with respect to the codebook $\boldsymbol{U}_k$ (Yu et al., 2023; Xu et al., 2025).

The compression module is derived from the update of *codes* $\boldsymbol{\alpha}_k$ as shown in Eq. (6). Given a token feature $\boldsymbol{z}$ and its *code* $\boldsymbol{\alpha}_k$ in $k$-th subspace supported by $\boldsymbol{U}_k$, we seek to update $\boldsymbol{\alpha}_k$ according to its memberships $\pi_k$ as shown in Eq. (7). Hence, we have the following *code* update:

$$\boldsymbol{\alpha}_k^{\ell+1} = \boldsymbol{\alpha}_k^{\ell} - \pi_k(\boldsymbol{I} - \boldsymbol{Q}_k\boldsymbol{Q}_k^T)\boldsymbol{\alpha}_k^{\ell} \in \mathbb{R}^p \tag{19}$$

We then transform the *codes* into token feature by the following formula:

$$z_{\ell+1} = \sum_{k=1}^{K} U_k \alpha_k^{\ell+1} \tag{20}$$

$$= \sum_{k=1}^{K} U_k (\alpha_k^\ell - \pi_k (I - Q_k Q_k^T) \alpha_k^\ell) \tag{21}$$

$$= \sum_{k=1}^{K} U_k [U_k^T z_\ell - \pi_k (I - Q_k Q_k^T) U_k^T z_\ell] \tag{22}$$

$$= \underbrace{\sum_{k=1}^{K} U_k U_k^T z_\ell}_{\approx z_\ell} - \sum_{k=1}^{K} \pi_k U_k (I - Q_k Q_k^T) U_k^T z_\ell \tag{23}$$

Based on the union of $K$ low-dimensional subspaces model mentioned above, where $z = U_k \alpha$, Eq. (20) can be interpreted as follows: $U_k \alpha_k$ obtained in each subspace represents the component of the original token $z$ in that subspace. Therefore, we can sum these components to recover the original $z$. The approximation in Eq. (23) can be understood in the same manner.

### B.3. Details on Obtaining the Basis of Column Space

To ensure the paper is self-contained, we briefly explain the steps of obtaining an orthogonal basis through Cholesky decomposition. For simplicity, we assume here that $Z$ has already captured the column space by multiplication with the random matrix $\Omega$. Given a Gram matrix $G = Z^T Z \in \mathbb{R}^{m \times m}$, we can obtain the lower triangular matrix $L$ via Cholesky decomposition $G = LL^T$. Hence, the lower triangular matrix $L$ can be used to calculate the orthogonal basis $Q$ in the following way. Consider that $QQ^T = I$ and we seek to find a coefficient matrix $Q_{\text{coef}}$ such that $Q = ZQ_{\text{coef}}$. Then we have

$$(ZQ_{\text{coef}})^T (ZQ_{\text{coef}}) = I \tag{24}$$

$$Q_{\text{coef}}^T (Z^T Z) Q_{\text{coef}} = I \tag{25}$$

$$Q_{\text{coef}}^T G Q_{\text{coef}} = I \tag{26}$$

Since we have $G = LL^T$, therefore

$$Q_{\text{coef}}^T L L^T Q_{\text{coef}} = I \tag{27}$$

$$(L^T Q_{\text{coef}})^T (L^T Q_{\text{coef}}) = I \tag{28}$$

Here, we can set $L^T Q_{\text{coef}} = I$, thereby this upper triangular linear system of equations can be solved by backward substitution.

$$Q_{\text{coef}} = (L^T)^{-1} \tag{29}$$

$$\Rightarrow Q = ZL^{-T} \tag{30}$$

### B.4. Detailed Analysis of Complexity

*Table 2.* Computational steps and complexity analysis

| Step | Expansion Module | | | Compression Module | | |
|---|---|---|---|---|---|---|
| | **Matrix Operation** | **Time** | **Space** | **Matrix Operation** | **Time** | **Space** |
| 1 | $Y = Z\Omega$ | $\mathcal{O}(dnKr)$ | $\mathcal{O}(nKr)$ | $Y_k = A_k\Omega$ | $\mathcal{O}(pnr)$ | $\mathcal{O}(pn)$ |
| 2 | $M = Y^T Y$ | $\mathcal{O}(dK^2r^2)$ | $\mathcal{O}(K^2r^2)$ | $M_k = Y^T Y$ | $\mathcal{O}(pr^2)$ | $\mathcal{O}(r^2)$ |
| 3 | $L = \text{ChoL}(M)$ | $\mathcal{O}(\frac{1}{3}K^3r^3)$ | $\mathcal{O}(K^2r^2)$ | $L_k = \text{ChoL}(M_k)$ | $\mathcal{O}(\frac{1}{3}r^3)$ | $\mathcal{O}(r^2)$ |
| 4 | $Q = YL^{-T}$ | $\mathcal{O}(dK^2r^2)$ | $\mathcal{O}(dKr)$ | $Q_k = Y_kL_k^{-T}$ | $\mathcal{O}(pr)$ | $\mathcal{O}(pr)$ |
| 5 | $I - QQ^T$ | $\mathcal{O}(d^2Kr)$ | $\mathcal{O}(d^2)$ | $I - Q_kQ_k^T$ | $\mathcal{O}(p^2r)$ | $\mathcal{O}(p^2)$ |
| **Total** | | $\mathcal{O}(dnKr)$ | $\mathcal{O}(nKr)$ | | $\mathcal{O}(pnr)$ | $\mathcal{O}(pn)$ |

# C. Experiment Details

## C.1. Practical Implementation

**Class attention layer for information aggregation.**   For supervised classification task, a typical way to aggregate class-related information is by inserting a learnable [CLS] token in each layer. However, since PACEAttention aims to compress different group features onto corresponding subspaces, inserting a [CLS] token at each layer may not be suitable as it do not belong to any subspaces, thereby cannot aggregate class-related information effectively. Hence, we use global class attention layer Ali et al. (2021) at the end of entire architecture.

**Numerical stability.**   Since Cholesky decomposition can only be applied to positive definite matrices, we add a small identity matrix $\epsilon\boldsymbol{I}$ to the Gram matrix $\boldsymbol{Y}^T\boldsymbol{Y}$ to ensure numerical stability. In fact, after incorporating the identity matrix, the resulting projection matrix $\boldsymbol{Q}\boldsymbol{Q}^T$ is equivalent to obtaining a *regularized version* of the projection matrix. This is analogous to the operators in ReduNet, which have strong connections to ridge regression.

**Relaxing orthonormality constraints.**   Similar to the previous methods (Yu et al., 2023; Wu et al., 2025), we do not enforce the orthonormality constraints of $\boldsymbol{U}_{[K]}$, as strict orthonormality would require computationally expensive manifold optimization. Besides, the key null-space projection is obtained via Cholesky decomposition where $\boldsymbol{Q}$ is orthonormal. Hence, the resulting core transformation is based on an orthonormalized structure.

## C.2. Model Configurations

*Table 3.* The configurations and parameters of PACENet

|  | **PACENet-T** | **PACENet-S** | **PACENet-M** | **PACENet-B** | **PACENet-B+** | **PACENet-L** |
|---|---|---|---|---|---|---|
| # parameters | 1.91M | 7.2M | 14.7M | 21.0M | 57.7M | 70M |
| # attention heads $K$ | 4 | 8 | 8 | 16 | 16 | 16 |
| # layers $L$ | 12 | 12 | 24 | 48 | 24 | 36 |
| # feature dimension $d$ | 192 | 384 | 512 | 512 | 1024 | 1024 |
| # head dimension $p$ | 48 | 48 | 64 | 64 | 64 | 64 |

*Table 4.* The configurations and parameters of ToST

|  | **ToST-T** | **ToST-S** | **ToST-M** |
|---|---|---|---|
| # parameters | 5.8M | 22.6M | 68.1M |
| # attention heads $K$ | 4 | 8 | 8 |
| # layers $L$ | 12 | 12 | 24 |
| # feature dimension $d$ | 192 | 384 | 512 |
| # head dimension $p$ | 48 | 48 | 64 |

*Table 5.* The configurations and parameters of CRATE

|  | **CRATE-Tiny** | **CRATE-Small** | **CRATE-Base** | **CRATE-Large** |
|---|---|---|---|---|
| # parameters | 6.1M | 13.1M | 22.8M | 77.6M |
| # attention heads $K$ | 6 | 12 | 12 | 16 |
| # layers $L$ | 12 | 12 | 12 | 24 |
| # feature dimension $d$ | 384 | 576 | 768 | 1024 |
| # head dimension | 32 | 48 | 64 | 64 |

Due to the quadratic time complexity of CRATE , as well as limited computational resources, our experiments primarily compare against ToST. For CRATE model, we directly use the results reported in its papers(Yu et al., 2023). As shown in

Table 3, we tested PACENet models with six different parameters scales including PACENet-T(Tiny), PACENet-S(Small), PACENet-M(Medium), PACENet-B(Base), PACENet-B+(Base+) and PACENet-L(Large). Note that under identical settings, our model actually has significantly fewer parameters than ToST as shown in Table 3 and Table 4.

### C.3. Training Setup

*Table 6.* Batch size settings for different PACENet model scales

| Model | Batch Size |
|---|---|
| PACENet-T | 2048 |
| PACENet-S | 1024 |
| PACENet-M | 512 |
| PACENet-B | 256 |
| PACENet-B+ | 256 |
| PACENet-L | 128 |

**Pre-training on ImageNet-1k.** Our models are trained using the AdamW optimizer with a learning rate of $5 \times 10^{-4}$ for 500 epochs. Due to computational resource constraints, we used different batch sizes for models of different scales as shown in Table 6. All images are resized to $224 \times 224$ resolution and divided into $16 \times 16$ patches, which are then converted into patch embeddings. For other training configurations, we follow the same protocol as in (Wu et al., 2025). All pre-training experiments are conducted on 2 NVIDIA H100 GPUS.

**Fine-tuning.** Fine-tuning experiments are conducted using pretrained PACENet and ToST models as initialization, then fine-tuning them on the following datasets: CIFAR-10/100 (Krizhevsky et al., 2009), Oxford Flowers (Nilsback & Zisserman, 2008), and Oxford-IIIT Pets (Parkhi et al., 2012). All fine-tuning experiments employ a batch size of 256 with a learning rate of $1 \times 10^{-4}$.

## D. Additional Experimental Studies

### D.1. Regulate Expansion–Compression Dynamics via Enforcing Weight Positivity

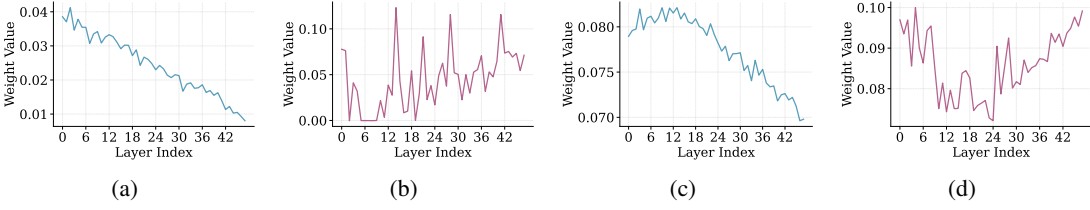

(a)  (b)  (c)  (d)

*Figure 11.* The weight distribution of PACENet-B after training on CIFAR-10. (a) Weight of the expansion module. (b) Weight of the compression module. (c) Weight of the expansion module after enforcing weight positivity. (d) Weight of the compression module after enforcing weight positivity.

As shown in Figure 11, after applying the softplus function (Dugas et al., 2000) to enforce weight positivity, the relative fluctuations of the weights across layers are reduced, as evidenced by the narrower vertical axes in Figures 11(c) and 11(d). Moreover, all layers are activated and contribute to feature updates, unlike in Figure 11(b), where some layers are inactive.

### D.2. Empirical Analysis of Capacity Allocation on CIFAR-100

As the Figure 12 shows, a substantial portion of ToST's parameters and performance gains stems from its MLP layers. Under comparable parameter budgets, PACENet-S achieves higher accuracy than ToST-S-w/o MLP.

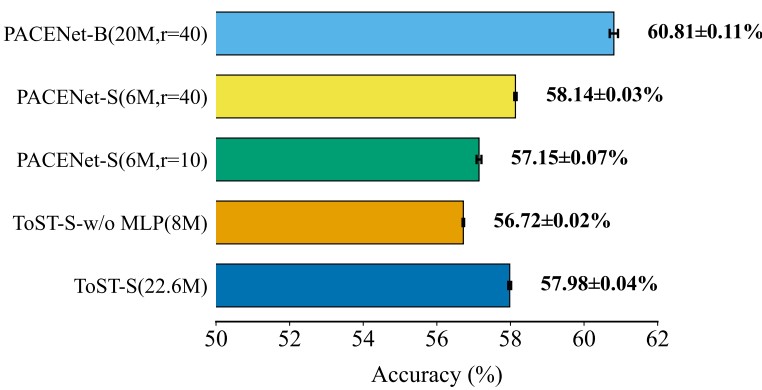

*Figure 12.* A fairer comparison with ToST on CIFAR-100 after 100 epochs training.

### D.3. Ablation Study on Expansion Module

We evaluate the impact of the expansion module using PACENet-S on CIFAR-10 over 400 training epochs. As shown in Table 7, the model with the expansion module exhibits better accuracy.

*Table 7.* Accuracy (%) with and without the expansion module.

| Dataset | w/o Expansion | w/ Expansion |
|---------|---------------|--------------|
| CIFAR-10 | 90.40 | **90.71** |

### D.4. Compare with CBT on ImageNet-1k

For completeness, we additionally include a comparison with CBSA/CBT (Wen et al., 2026) on ImageNet-1K. Note that the reported CBT results are taken from the original paper and are intended only as a rough reference rather than a strictly controlled comparison.

*Table 8.* Compare with CBT on ImageNet-1k

|  | PACENet-T | PACENet-S | PACENet-B | PACENet-B+ |
|---|-----------|-----------|-----------|------------|
| # Parameters | 1.91M | 7.2M | 21.0M | 57.7M |
| Top-1 Acc | 54.1 | 67.8 | 74.8 | 75.5 |
|  | **CBT-T** | **CBT-S** | **CBT-B** | **CBT-L** |
| # Parameters | 1.8M | 6.7M | 25.7M | 83.1M |
| Top-1 Acc | 63.2 | 71.4 | 73.4 | 74.4 |

### D.5. Two Perspectives on Feature Compression: Column-space Projection vs. Null-space Subtraction

We conduct this experiment on toy data, as shown in Figure 13(b), compression via subtracting the null space projection enables smoother optimization. In contrast, directly projecting onto the column space is not effective, either in terms of layer-wise compression or multi-epoch optimization.

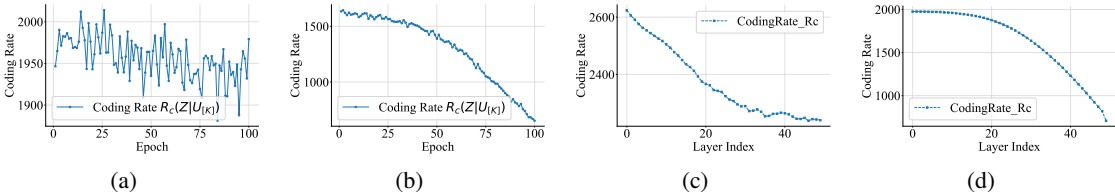

*Figure 13.* (a) Compression via directly projecting onto the column space. (b) Compression via subtracting the null space projection. (c) Layer-wise compression via directly projecting onto the column space. (d) Layer-wise compression via subtracting the null space projection.

## D.6. Sensitivity Experiments on Regularization Term in Cholesky Decomposition.

*Table 9.* Ablation study on the regularization term $\epsilon$. We report the results after 40 training epochs on the CIFAR-10 dataset.

| $\epsilon$ | Time/Epoch (m:s) | Fail Rate (%) | Acc (%) |
|---|---|---|---|
| 0 | - | - | - |
| $10^{-4}$ | 8:12 | 21.28 | 50.11 |
| $10^{-3}$ | 7:33 | 11.11 | 71.96 |
| $10^{-2}$ | 0:55 | 0 | 71.80 |
| $10^{-1}$ | 0:55 | 0 | 71.59 |

A common trick to ensure numerical stability during Cholesky decomposition is to add a small identity matrix term $\epsilon\mathbf{I}$ to the diagonal. As shown in Table 9, we evaluate the failure rate of Cholesky decomposition (i.e., when it falls back to the more expensive QR decomposition), the corresponding Top-1 accuracy, and the per-epoch training time under different values of $\epsilon$. When $\epsilon = 10^{-2}$, Cholesky decomposition never fails, while delivering both reasonable training speed and accuracy. This is therefore adopted as the default setting in our model. In contrast, setting $\epsilon = 0$ (i.e., no stabilization) quickly leads to NaN values during training, making the model untrainable. For smaller values such as $\epsilon = 10^{-3}$ and $\epsilon = 10^{-4}$, we observe that the failure probability of Cholesky decomposition significantly increases. Consequently, the training process frequently falls back to QR decomposition, resulting in a substantial increase in per-epoch training time.

## D.7. More Attention Visualizations

Here, we show visualizations of attention from more layers. Due to layer-wise compression, deeper layers may exhibit fragmented attention. Nevertheless, different heads in each layer still focus on distinct parts of the image.

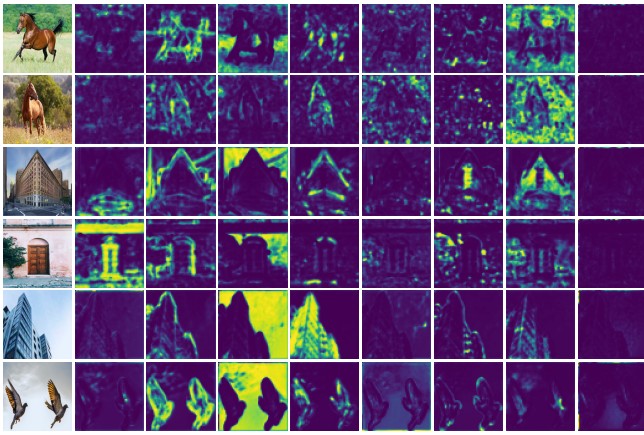

*Figure 14.* **Membership distribution $\pi$ of PACENet-B. Results are shown for layer 2.**

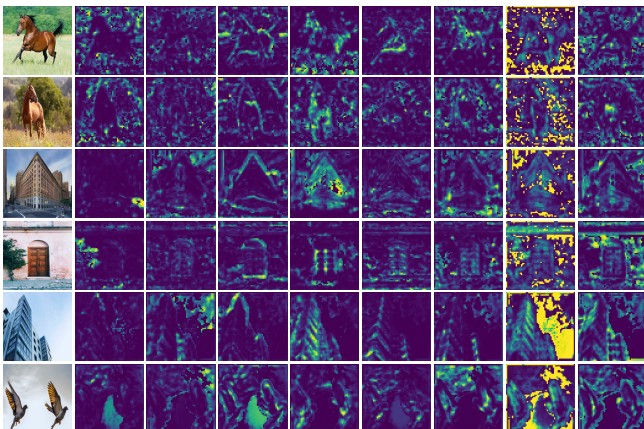

*Figure 15.* **Membership distribution $\pi$ of PACENet-B. Results are shown for layer 20.**

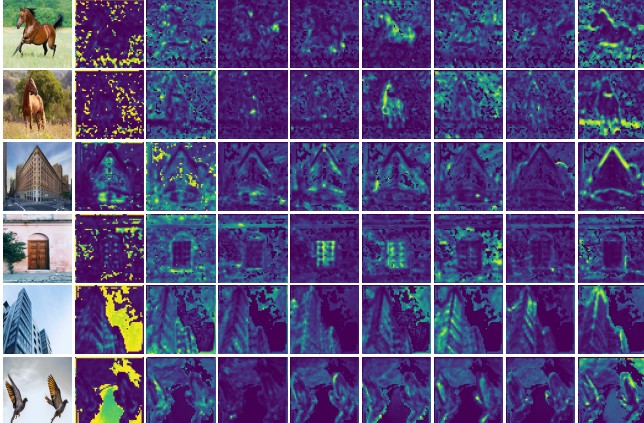

*Figure 16.* **Membership distribution $\pi$ of PACENet-B. Results are shown for layer 40.**

## E. Pseudocode of PACEAttention

See Algorithms 1, 2, 3 and 4 on the following pages.

## F. Use of Large Language Models

We declare that large language models (LLMs) were used solely for language polishing in this paper. All ideas, methods, experiments, and conclusions are the authors' own, and no substantive content was generated by LLMs.

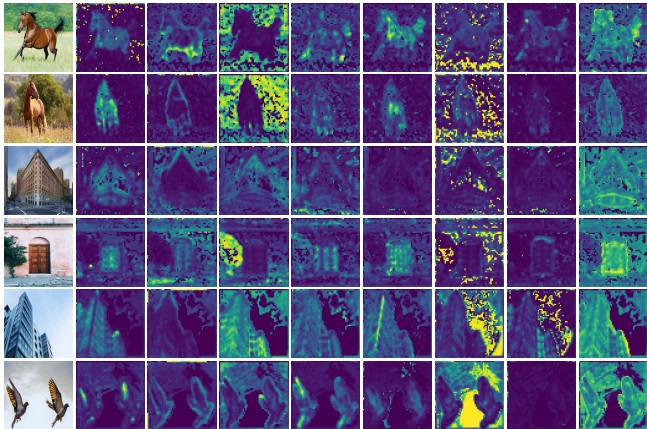

*Figure 17.* **Membership distribution $\pi$ of PACENet-B. Results are shown for layer 48.**

---

**Algorithm 1** Expansion module of PACEAttention in PyTorch

---

```python
class Expand(nn.Module):
    def __init__(self, dim, heads, subspace_rank):
        super().__init__()
        self.dim = dim
        self.heads = heads
        self.rank = subspace_rank * heads
        # Random matrix for capturing column space
        self.register_buffer(
            'Omega', torch.randn(n_tokens, self.rank)
        )

    def forward(self, x):
        b, n, d = x.shape
        x_t = x.transpose(-1, -2)

        # capturing column space
        Y = torch.matmul(x_t, self.Omega)

        # Obtaining the basis of Y via Cholesky decomposition
        Q = cholesky_orthogonalization(Y)

        # Null space projection
        QQT = torch.matmul(Q, Q.transpose(-1, -2))
        null_proj = x_t - torch.matmul(QQT, x_t)

        return null_proj.transpose(-1, -2)
```

---

**Algorithm 2** Compression module of PACEAttention in PyTorch

```python
class Compress(nn.Module):
    def __init__(self, dim, heads, dim_head, subspace_rank):
        super().__init__()
        self.heads = heads
        self.subspace_rank = subspace_rank
        self.Us = nn.Parameter(torch.randn(dim, dim_head * heads))
        self.temperature = nn.Parameter(torch.ones(1))
        # Random matrix for capturing column space
        self.register_buffer(
            'Omega', torch.randn(n_tokens, subspace_rank)
        )

    def forward(self, x):
        b, n, d = x.shape
        x_t = x.transpose(-1, -2)

        Us_T = rearrange(self.Us, 'd (h p) -> h p d', h=self.heads)
        alphas = torch.einsum('hpd,bdn->bhpn', Us_T, x_t)

        # Per-head null space projection
        Qs = []
        for i in range(self.heads):
            Y_i = torch.einsum(
                'bpn,nc->bpc', alphas[:, i], self.Omega
            )
            Q_i = cholesky_orthogonalization(Y_i)
            Qs.append(Q_i)

        Qs = torch.stack(Qs, dim=1)
        QQT = torch.matmul(Qs, Qs.transpose(-1, -2))

        # Column space and null space projections
        col_proj = torch.einsum('bhpp,bhpn->bhpn', QQT, alphas)
        null_proj = alphas - col_proj

        # Attention-based weighting
        norms = torch.linalg.norm(col_proj, dim=-2)
        weights = F.softmax(norms / self.temperature, dim=-1)

        # Weighted aggregation and reconstruction
        w_null_proj = torch.einsum(
            'bnh,bhpn->bhpn', weights, null_proj
        )
        w_null_proj_re = rearrange(
            w_null_proj, 'b h p n -> b (h p) n'
        )
        output = torch.matmul(self.Us, w_null_proj_re)

        return output.transpose(-1, -2)
```

**Algorithm 3** PACEAttention in PyTorch

```python
class PACEAttention(nn.Module):
    def __init__(self, dim, num_heads, subspace_rank):
        super().__init__()
        self.expand = Expand(dim, num_heads, subspace_rank)
        self.compress = Compress(
            dim, num_heads, dim//num_heads, subspace_rank
        )

        self.gamma1_raw = Parameter(
            log(exp(tensor(eta1)) - 1.0)
        )
        self.gamma2_raw = Parameter(
            log(exp(tensor(eta2)) - 1.0)
        )

    # Expand coefficient
    @property
    def alpha(self):
        return softplus(self.gamma1_raw)

    # Compress coefficient
    @property
    def beta(self):
        return softplus(self.gamma2_raw)

    def forward(self, x):
        # x: (batch, n_tokens, dim)
        expand_out = self.expand(x)
        compress_out = self.compress(x)

        # Residual connection with learnable coefficients
        x = x + self.alpha * expand_out - self.beta * compress_out
        return x
```

---

**Algorithm 4** Cholesky decomposition step in PyTorch

---

```python
def cholesky_orthogonalization(Y, eps=0.01):
    """
    Obtaining the basis of the column space using Cholesky
    decomposition.

    Args:
        Y: Input matrix of shape (batch, dim, rank)
        eps: Regularization parameter for numerical stability

    Returns:
        Q: Orthogonalized matrix of shape (batch, dim, rank)
    """
    # Normalize columns for numerical stability
    Y_norm = F.normalize(Y, p=2, dim=-2)

    try:
        # Compute Gram matrix: G = Y_norm^T @ Y_norm
        G = torch.matmul(Y_norm.transpose(-1, -2), Y_norm)

        # Add regularization: G = G + eps * I
        I = torch.eye(G.size(-1), device=G.device)
        G_reg = G + eps * I

        # Cholesky decomposition: G_reg = L @ L^T
        L = torch.linalg.cholesky(G_reg)

        # Solve triangular system: L^T @ C = I
        C = torch.linalg.solve_triangular(
            L.transpose(-1, -2), I, upper=True
        )

        # Compute orthogonal matrix: Q = Y_norm @ C
        Q = torch.matmul(Y_norm, C)

    except RuntimeError:
        # Fallback to QR decomposition if Cholesky fails
        Q, _ = torch.linalg.qr(Y_norm, mode="reduced")

    return Q
```

---

