# OpenReview forum: "PACEAttention: Principled and Adaptive Feature Compression-Expansion Grounded in the Geometry of $\text{MCR}^2$"
_ICML.cc/2026/Conference — ICML 2026 regular_

### Official Review · Reviewer_1e5Z · 2026-02-18

**Soundness:** 3
**Presentation:** 2
**Significance:** 2
**Originality:** 3
**Overall Recommendation:** 4
**Confidence:** 4

**Summary:**

The paper adopts an expansion-and-compression objective named MCR2 as the underlying optimization objective behind the proposed interpretable Transformer model, PACENet. The authors conduct extensive experiments to verify their design and interpretations, demonstrate a promising result for principled architecture research.

Specifically, the paper employs an estimated orthogonal basis, $Q$, to guide the updates for expansion and compression, respectively.
The features are expanded in the ambient space along the null space of $Q$, and compressed along the column space of $Q_k$.
The resulting interpretable attention mechanism, PACEAttention, are thus divided into two branches: expansion module and compression module.

Moreover, the paper investigated the balance between expansion and compression, enabled by the learned parameters $\alpha$ and $\beta$, and finds that the deeper layers of PACENet is mainly dominated by the compression term. Compared to ToST, (1) the segmentation property of PACENet is more pronounced; (2) the fairly compared top-1 accuracy is improved on CIFAR-10 while retaining linear complexity.

**Compliance With Llm Reviewing Policy:**

Affirmed.

**Final Justification:**

During the rebuttal, the authors provided a satisfactory discussion of related work and promised to include a broader comparison in the final version.

**Key Questions For Authors:**

Questions:

1. Do the authors plan to release their code and models for reproducibility, if the paper is accepted?

2. Is generating random projection matrices to estimate $Q$ kind of burdensome and expensive under common Pytorch implementations?

3. What is the fundamental difference between the formulation of PACEAttention and CBSA? Can PACEAttention be integrated into the unified formulation proposed in [2]?

4. Is it feasible to build a hybrid white-box model that mixes MSSA, TSSA, CBSA and PACEAttention across layers to trade off accuracy, speed, and interpretability?

**Limitations:**

See the Strengths And Weaknesses part.

**Strengths And Weaknesses:**

Strengths:

1. **Principled Architecture Design:** The reviewer appreciates the key insight of the paper very much (as illustrated by Figure 1b). The introduction of $Q$ makes the optimization of MCR2 and the resulting unrolled module elegant and intuitive. This is clear improvement upon ToST/TSSA, unlocking a complicated interplay among channels (i.e., dimensions) without the variational formulation.

2. **In-depth Experiments:** The experiments measuring the dynamics (or evolution) of layer-wise coding rate with respect to expansion and compression are interesting and reveal non-trivial findings of how PACENet echoes with its designing principle, MCR2.

3. **Fair Writing and Presentation:** The motivation, key designs, contributions are generally well presented and organized throughout the paper. However, as discussed in the weaknesses section, this is still not fully satisfactory: the paper fails to give a high-level overview of the field (e.g., a meaningful discussion of related work and a clear explanation of how PACEAttention improves upon prior methods).

4. **Specially Appreciated Points:** The orthogonal basis is mathematically estimated via random projection, which is a nice try to approximate the principal subspace in deep models. Thanks to $Q$, PACEAttention explicitly models dynamic features directions to be suppressed, as well as those for expansion.

Weaknesses:

1. **(Major) Insufficient Related Work and Ignorance of More Recent Advances**: The reviewer recognized that this paper is mainly built upon ToST (ICLR 2025). However, the introduction of a smaller set of tokens (i.e., $Q$) in MCR2 has been tried in DEPICT [1] (NeurIPS 2024) for semantic segmentation, where $Q$ is interpreted as principal directions and spans the principal subspace. More recently, CBSA [2] (NeurIPS 2025) has leveraged $Q$ as representative tokens, which is a more relaxed assumption and thereby unified both MSSA and TSSA. The reviewer believes that PACEAttention shares a similar yet distinct motivation and design to [1-4]; the authors should give a comprehensive discussion and comparisons in the paper (e.g. in a related work section).

2. **(Major) Unfair Setting for Comparison**:
The accuracy of CRATE/MSSA reported in Table 1 follows the results from the original paper, where CRATEs are not trained very well (e.g., using linear embedding instead of convolutional embedding). Intuitively, the performance of PACEAttention should be better than ToST, but remain inferior to MSSA. The reviewer also wonder the performance comparisons with CBSA [2].

3. **(Minor) Image Classification Only**: The authors are suggested to involve experiments on more tasks, e.g. dense prediction, which has been mentioned in the Limitations and Future Works section.

In summary, the paper is around the publication borderline for a top-tier conference, but it is significantly weakened by (1) the lack of a comprehensive survey and (2) inconsistent baseline comparisons on ImageNet-1K.
In my opinion, weaker performance can be acceptable in this research direction, provided that the paper offers good insights and discussion (i.e., doing it right before doing it good).

Therefore, the reviewer would like to give **weak reject** in this reviewing stage.
The reviewer is willing to raise the recommendation to **weak accept** if the authors address the two major weaknesses above.


[1] Rethinking decoders for transformer-based semantic segmentation: A compression perspective, NeurIPS 2024.

[2] Towards Interpretable and Efficient Attention: Compressing All by Contracting a Few, NeurIPS 2025.

[3] Attention-only transformers via unrolled subspace denoising, ICML 2025.

[4] A Nyström-Based Algorithm for Approximating Self-Attention, AAAI 2021.

---

> ### Author Rebuttal · Authors · 2026-03-28
>
> **Q1: release code?**
>
> **Response:**  Yes. If the paper is accepted, we would be happy to release our code.
>
> **Q2: Is generating random projection matrices to estimate $Q$ kind of burdensome and expensive under common Pytorch implementations?**
>
> **Response:** As shown in Figure 2, our model achieves linear complexity. Besides, during training, the random matrices used in each layer are generated only once (e.g., at the first batch of the first epoch) and reused thereafter. Hence, they do not introduce additional overhead during training.
>
> Moreover, the use of randomized projection not only captures the column space but also effectively reduces the dimensionality. As a result, the subsequent Cholesky decomposition is performed in a reduced space, leading to a constant computational complexity of $O(1/3r^3)$, where $r = 20$. A detailed analysis is provided in Appendix B.4.
>
> **Q3:The fundamental difference between PACEAttention and CBSA? Can PACEAttention be integrated into the unified formulation proposed in [2]?**
>
> **Response:**
> (**Major issue 1**)
> CBSA provides an elegant unified framework for attention.
> While both PACEAttention and CBSA reduce computational complexity, they differ fundamentally: PACEAttention exploits low-dimensional structure, whereas CBSA selects representative tokens. Therefore, our main contribution lies in approaching this problem from a geometric perspective.
> We hope that this viewpoint can provide useful insights to the community, and we will include a discussion of related work in the final version.
>
> Moreover, we observe that several models, including CBSA [1], DEPICT [2], and MSSA [3], can be written in the form $UU^T Z S((U^T Z)^T (U^T Z))$, where $S(\cdot)$ denotes softmax. As discussed in lines 131–136, this formulation may deviate from the intended compression objective, since MSSA is derived from a second-order approximation with the first-order term omitted [4]. Even when the first-order term is retained, Fig. 1(b) in [4] suggests that the eigenvalues of $(I + (U^T Z)^T (U^T Z))$ must be less than 2, which is difficult to guarantee in practice. Therefore, this remains an open direction for further investigation.
>
> Regarding the possibility of unifying PACEAttention with the CBSA formulation, we believe this can be better understood by examining its relationship to MSSA. Our method updates features using the low-dimensional structure of $U_k^T Z$, denoted as $Q_k$. If we instead discard $Q_k$ and use $((U^T Z)^T (U^T Z))$ to guide feature updates, the formulation becomes closely related to the softmax module in MSSA.
>
> (**Major issue 2**) We attempted to train CRATE/MSSA for a direct comparison. However, due to its quadratic complexity and our limited computational resources, we were unable to complete large-scale experiments.
>
> Compared to the results reported in CBSA/CBT [1] paper,
> our method achieves slightly better performance than CBSA. For example, on ImageNet-1K, CBT-L (83.1M) reports 74.4% top-1 accuracy, while our model PACENet-B+ (57.7M) achieves 75.6%.
>
> Moreover, as reported in Table 4 of the CBSA paper, when MSSA is combined with convolutional embedding layers, its performance is slightly better than CBSA. This suggests that our method may achieve competitive or better performance under similar settings.
> Due to time constraints, we are unable to conduct a comprehensive comparison with CBSA at this stage. We will further investigate and verify this in future work.
>
> **Q4: Is it feasible to build a hybrid white-box model that mixes MSSA, TSSA, CBSA and PACEAttention across layers to trade off accuracy, speed, and interpretability?**
>
> **Response:** Briefly, TSSA can be viewed as an overly simplified variant of MSSA.
> CBSA focuses on selecting representative tokens, PACEAttention leverages low-dimensional structures to guide feature updates. Both methods aim to balance performance and efficiency through different forms of information compression.
>
> From a broader perspective, while hybrid approaches are certainly possible, a more principled and elegant direction may lie in unifying low-dimensional structures with the selection of high-quality samples under a common objective and architecture.
>
> **Reference:**
>
> [1] Towards Interpretable and Efficient Attention: Compressing All by Contracting a Few, NeurIPS 2025.
>
> [2] Rethinking decoders for transformer-based semantic segmentation: A compression perspective, NeurIPS 2024.
>
> [3] Yu, Yaodong, et al. "White-box transformers via sparse rate reduction: Compression is all there is?." *Journal of Machine Learning Research* 25.300 (2024): 1-128.
>
> [4] Hu, Yunzhe, Difan Zou, and Dong Xu. "An in-depth investigation of sparse rate reduction in transformer-like models." Advances in Neural Information Processing Systems 37 (2024): 116815-116837.

---

> > ### Author Rebuttal · Reviewer_1e5Z · 2026-04-02
> >
> > The authors’ response addressed most of my concerns, and I am happy to raise my rating. This paper tackles a promising problem, and I hope the work will have broader impact in the future.

---

> > > ### Author Response · Authors · 2026-04-02
> > >
> > > Thank you so much for confirming that our response has addressed your concerns.

---

### Official Review · Reviewer_sJvj · 2026-03-06

**Soundness:** 4
**Presentation:** 3
**Significance:** 3
**Originality:** 3
**Overall Recommendation:** 4
**Confidence:** 5

**Summary:**

The authors propose PACEAttention, a novel attention-like transformer block, and PACENet, a novel sequence-to-sequence neural network architecture built from iterating PACEAttention (+ residual connection and pre-layer-norms). They motivate PACEAttention as being derived from the principle of MCR$^{2}$, which says that features ought to be compressed to a low-rank structure (union of subspaces), which in this case is encoded by the weights, while expanded on each subspace. In order to develop neural network operators which naturally promote these kinds of features, they use random matrices and a low-dimensional Cholesky decomposition to rapidly compute projections onto the column space of the data's low-rank structure. Experiments demonstrate performance relative to other MCR$^{2}$-inspired networks, efficiency, and the level of mathematical/semantic interpretability.

**Compliance With Llm Reviewing Policy:**

Affirmed.

**Final Justification:**

My initial review was tentatively favorable because I believed some of the claims were significant enough to clear a bar (a new novel efficient architecture with some principles and favorable empirics), but believed some things were overstated. The rebuttal did not change this point: their "more principled design" still relies on some empirical tricks (the same weakness they point out in prior work) and reasoning by analogy instead of mathematics, and the empirical performance remains questionable if one removes these tricks. Hence, I am on the border and do not feel strongly either way, but am tentatively in favor of acceptance due to the core contribution being potentially interesting.

**Key Questions For Authors:**

(1 --- important for evaluation) Is there a way to interpret the PACEAttention as an optimization step on an MCR$^{2}$-like objective?
(2 --- clarification) The randomization step seems important and relevant to the stability. Do you have any experiments (even at synthetic scale or CIFAR10 would probably be enough signal) where you use a deterministic way to compute the column space and check that it reaches similar performance? If this deterministic method is worse, could you explain why?
(3 --- clarification) Are there any results for language tasks?

**Limitations:**

yes

**Strengths And Weaknesses:**

**Strengths:**
- The empirical study is very thorough, at least about ablations to the proposed architecture.
- The idea is nice; using randomization to compute column-spaces on the fly for "white-box" networks has not been done before. It is also very interesting that such a procedure is stable.
- The empirical results are quite strong, performing very well on ImageNet classification and transfer learning setups, compared to other "white box" models of similar numbers of parameters. The mathematical/semantic interpretability and efficiency are nice wins, too.

**Weaknesses**
- Despite claiming to be more principled than alternatives, PACE still uses heuristics; for example, the previous works [2, 3] framed attention as an optimization step on a particular objective up to some approximation, meanwhile PACEAttention does not claim to use such a mathematical derivation at all, and thus is arguably less principled. The "softplus hack" to align the behavior with theoretical predictions also contributes. In particular, this means that fair comparisons against ToST with MLP should not be excluded (nor the regular transformer, to be honest).
- There is some uncertainty about how the MLP-free PACENet architecture can scale, since attention-free architectures tend to have worse scaling curves [4], but the experiments at ImageNet scale are good.
- The work fixes an extra rank parameter $r$, claims it is constant in the computational model, and thus the forward pass which takes _cubic_ time in $r$ is asserted to only take constant time. There is no motivation for $r < p$; this design choice is not clear from [1, 2] for example; the features in each group should be "maximally expanded". In this case the complexity of the forward pass would be $p^{3}$ or worse.

[1]: Yu, Yaodong, et al. "Learning diverse and discriminative representations via the principle of maximal coding rate reduction." Advances in neural information processing systems 33 (2020): 9422-9434.
[2]: Yu, Yaodong, et al. "White-box transformers via sparse rate reduction." Advances in Neural Information Processing Systems 36 (2023): 9422-9457.
[3]: Wu, Ziyang, et al. "Token statistics transformer: Linear-time attention via variational rate reduction." arXiv preprint arXiv:2412.17810 (2024).
[4]: Wang, Peng, et al. "Attention-only transformers via unrolled subspace denoising." arXiv preprint arXiv:2506.03790 (2025).

---

> ### Author Rebuttal · Authors · 2026-03-28
>
> **Q1: Is there a way to interpret the PACEAttention as an optimization step on an MCR-like objective?**
>
> **Response:**
> As illustrated in Figure 1b, the gradient of MCR-like objective inherently performs compression and expansion in the feature space. Our PACEAttention is designed based on this geometric behavior.
> Hence, PACEAttention can be interpreted as a gradient step of an MCR^2-like objective.
>
>
> **Q2: The randomization step seems important and relevant to the stability. Do you have any experiments (even at synthetic scale or CIFAR10 would probably be enough signal) where you use a deterministic way to compute the column space and check that it reaches similar performance? If this deterministic method is worse, could you explain why?**
>
> **Response:** The use of a randomized method is not only for obtaining the column space, but also for computational efficiency. While deterministic approaches (e.g., SVD) are theoretically appealing, they are often computationally expensive for large matrices. To the best of our knowledge, there is no deterministic method that can compute the column space efficiently at large scale.
>
> In our case, after deriving the geometric interpretation of the objective gradient, we require a practical way to approximate the column space efficiently rather than exactly. The randomized approach provides a favorable trade-off between accuracy and efficiency, which makes it well-suited for our setting [1].
>
> **Q3: Are there any results for language tasks?**
>
> **Response:** We have not yet evaluated our method on language tasks. In future work, we plan to extend our study to both vision and language domains, with a particular focus on analyzing the characteristics of the learned representations.
>
> **Weaknesses: Motivation for setting of r**
>
> Regarding the choice of r, please refer to our response to Q1 from Reviewer KEqD.
>
> **Reference:**
>
> [1] Halko, Nathan, Per-Gunnar Martinsson, and Joel A. Tropp. "Finding structure with randomness: Probabilistic algorithms for constructing approximate matrix decompositions." SIAM review 53.2 (2011): 217-288.

---

> > ### Author Rebuttal · Reviewer_sJvj · 2026-04-03
> >
> > For Q1: I was wondering what is the actual objective for which the layer is approximately an optimization step, and in what sense does this "approximately" hold (e.g., what kind of insight can you obtain about the operator from the objective it is learning). Best if it were e.g. actual gradient descent on a geometrically interpretable modification of the rate reduction (such as eg ToST [1]). As is, the connection seems a bit non-rigorous (for example the definition of $r$).
> >
> > For Q2: The question is about whether using deterministic methods actually make the performance worse. I agree that randomized methods are better for efficiency in this case. But on small synthetic data or even MNIST you could, if you liked, compare the performance of small models with both methods. That comparison is what I was asking about.
> >
> > [1] Wu, Ziyang, et al. "Token statistics transformer: Linear-time attention via variational rate reduction." arXiv preprint arXiv:2412.17810 (2024).

---

> > > ### Author Response · Authors · 2026-04-04
> > >
> > > Sorry for the misunderstanding earlier, and thank you for your insightful comments.
> > > We hope the following response helps clarify and address your concerns.
> > >
> > > **For Q1:**
> > > At the beginning of Section 3 (lines 162--164), we clarify that the expansion and compression modules in our model, based on null-space projection, is derived from the **optimization objectives** $R(Z)$ in Eq. (1) and $R_c(Z \mid U_{[K]})$ in Eq. (2). These correspond to the expansion term in ReduNet [1] and the compression term in CRATE [2], respectively.
> > > Geometrically,
> > > the expansion term measures the volume(or coding rate) of the entire spanned space (i.e., column space) of features, and the compression term sum up the coding rate of token feautures on each subspace[2].
> > > Changes in coding rate can be regarded as changes in the size of the feature space. As shown in Figure 3(a), our expansion and compression modules can adjust the coding rate (i.e., the space size).
> > >
> > > For the expansion objective $R(Z)$ in Eq. (1), as shown in the ReduNet paper (see Remark 2 in [1]), the gradient of the expansion term  $R(Z)$ can be directly interpreted as ridge regression.
> > > Since ridge regression is the regularized version of the least square problem, the **gradient of $R(Z)$ can be geometrically understood as projecting features toward the null space** of the current feature representations [1,3]. Adding such a null-space component to a feature $z$ leads to an expansion effect, as illustrated in Fig. 1(b).
> > > Different from the expansion objective, the compression objective $R_c(Z \mid U_{[K]})$ in Eq. (2) is formulated in terms of the codes of features on the corresponding subspaces, namely $A_k = U_k^\top Z$. Nevertheless, its geometric interpretation remains analogous to that of the expansion objective.
> > >
> > > Therefore, we can interpret PACEAttention as approximating the geometric behavior induced by the gradients of $R(Z)$ and $R_c(Z \mid U_{[K]})$. **Here, the term "approximately" mainly refers to the practical construction of the null-space projection, which requires choosing the parameter $r$ to specify the retained dimension of the column space.** Concretely, the null-space projection is represented through a basis $Q$ of the column space, and $r$ is chosen empirically to determine how many column-space dimensions are retained in this approximation.
> > >
> > > **For Q2:**
> > > We conducted a quick verification of the deterministic approach on a small subset of the MNIST dataset (5,000 training samples / 1,000 test samples) using an A100 GPU. Specifically, we adopted an SVD-based method to obtain the column space of the features, and selected the top 20 components as $Q$, which is consistent with our choice of $r=20$.
> > >
> > > We compare the two methods in terms of runtime, accuracy, and convergence speed.
> > >
> > > First, in terms of runtime, the random method only requires 1 second per epoch, whereas the SVD-based method requires 91 seconds.
> > >
> > > Second, in terms of accuracy and convergence speed, we observe that the **random method performs better than the deterministic one**. Within 50 epochs, the random method achieves a top-1 accuracy of 96%, while the SVD-based method reaches 89.89%.
> > >
> > > One possible reason is that optimization through SVD itself can be numerically unstable, as also discussed in the official PyTorch documentation [4]. In addition, at the early stage of training, explicitly forcing the updates to follow the top-20 SVD directions may be overly "hard". By contrast, the random method is "softer", and this randomness may also provide a regularization effect that leads to better generalization.
> > >
> > >
> > >
> > > **Reference:**
> > >
> > > [1]Chan, Kwan Ho Ryan, et al. "Redunet: A white-box deep network from the principle of maximizing rate reduction." Journal of machine learning research 23.114 (2022): 1-103.
> > >
> > > [2]Yu, Yaodong, et al. "White-box transformers via sparse rate reduction." Advances in Neural Information Processing Systems 36 (2023): 9422-9457.
> > >
> > > [3]van Wieringen, Wessel N. "Lecture notes on ridge regression." arXiv preprint arXiv:1509.09169 (2015).
> > >
> > > [4] https://docs.pytorch.org/docs/stable/generated/torch.linalg.svd.html

---

### Official Review · Reviewer_KEqD · 2026-03-12

**Soundness:** 3
**Presentation:** 4
**Significance:** 3
**Originality:** 3
**Overall Recommendation:** 4
**Confidence:** 4

**Summary:**

This paper introduces PACEAttention, a principled and adaptive attention mechanism grounded in the geometric interpretation of the Maximal Coding Rate Reduction (MCR2) objective. The authors identify a gap between existing white-box models (like CRATE) and the actual gradients of the MCR2 objective, proposing a novel framework that utilizes null space and column space projections to explicitly perform feature expansion and compression. By integrating randomized subspace identification and Cholesky-based orthogonalization, the authors achieve linear computational complexity O(n) relative to the sequence length. The resulting PACENet demonstrates superior efficiency and performance, achieving 79.0% Top-1 accuracy on ImageNet-1K with only 7.2M parameters, significantly outperforming prior principled architectures in both scale and accuracy.

**Compliance With Llm Reviewing Policy:**

Affirmed.

**Ethical Review Concerns:**

NA.

**Final Justification:**

Based on my review and the authors' rebuttal, which partially addresses my concerns, I would like to keep my recommendation to this paper.

**Key Questions For Authors:**

1. How does the optimal choice of r vary across different layers or datasets, and is there a principled way to determine r?
2. Given the sensitivity of Cholesky decomposition to numerical precision, what specific epsilon values or regularization techniques are necessary to ensure stability during FP16 or BF16 training?

**Limitations:**

Yes. The authors acknowledge that while PACEAttention is more interpretable and efficient than previous white-box models, it still trails the most highly-engineered black-box SOTA models in pure accuracy, and they provide an honest assessment of the current architectural constraints.

**Strengths And Weaknesses:**

This paper is technically sound and highly original, offering a rigorous geometric derivation that links MCR2 gradients to subspace projections. The introduction of "null space subtraction" for feature expansion is a theoretically grounded contribution that enhances the interpretability of the attention mechanism. In terms of significance, the achievement of linear complexity addresses a critical bottleneck for white-box Transformers, making them viable for large-scale tasks. The presentation of this paper is professional, supported by both synthetic "toy" experiments and large-scale benchmarks. However, a primary weakness lies in the limited comparison with mainstream black-box models like Swin Transformer or DeiT regarding training stability and convergence speed under standard recipes. Additionally, the sensitivity of the pre-defined rank r is not fully explored across different dataset complexities, and several crucial implementation details regarding the randomized algorithms are relegated to the appendix, which slightly hinders immediate reproducibility from the main text.

---

> ### Author Rebuttal · Authors · 2026-03-28
>
> **Q1: How does the optimal choice of r vary across different layers or datasets, and is there a principled way to determine r?**
>
> **Response:** As noted in line 207 of the manuscript, a basic principle is that the dimension of the overall feature column space, $Kr$, should not exceed the feature dimension $d$ , where $K$  is the number of heads. This implies that the setting of r, must satisfy $r \leq p = d / K$ , where $p$  is the dimension of each head.
>
>
> Based on our experiments, as well as considerations of computational cost and accuracy, we set r = 20 for ImageNet experiments with the following two reasons.
> First, Pope et al. [1] measure the intrinsic dimensionality across multiple image datasets, including MNIST, CIFAR, and ImageNet. For a dataset like ImageNet, the intrinsic dimensionality is estimated to be in the range of 26 to 43. Therefore, as shown in Figure 8b of the manuscript, in our small-scale experiments, we accordingly set r to values such as 10, 20, 30, and 40, and observe that increasing r beyond this range yields limited additional gains.
>
> Second, although we set r = 20 uniformly across all layers in our current model, we further analyze the singular value spectrum of the learned features at different depths. We observe that, as the depth increases, the feature energy becomes increasingly concentrated in the top-20 dimensions. Specifically, the top-20 components account for 89.5% of the total energy at the first layer, 97.97% at the 20th layer, and 99.3% at the final layer. This observation provides empirical justification for our choice of r = 20, indicating that it is sufficient to capture the dominant feature subspace.
> We will further clarify this in the final version.
>
>
> **Q2: Given the sensitivity of Cholesky decomposition to numerical precision, what specific epsilon values or regularization techniques are necessary to ensure stability during FP16 or BF16 training?**
>
> **Response:** In our implementation, although we adopt mixed-precision training via `torch.autocast`, the Cholesky decomposition is explicitly performed in FP32 by casting the input as `Y = Y.float()` before decomposition. Therefore, the numerically sensitive operation is not carried out in FP16/BF16, which ensures stability.
>
> We provide an ablation study on the choice of epsilon in Appendix D.5, which shows that epsilon = 10^{-2} achieves a good balance between numerical stability and performance. Depending on the acceptance, we will release our code to facilitate reproducibility and further verification.
>
> **Reference:**
>
> [1] Pope, Phil, et al. "The Intrinsic Dimension of Images and Its Impact on Learning." *International Conference on Learning Representations*.

---

> > ### Author Rebuttal · Reviewer_KEqD · 2026-04-05
> >
> > I appreciate the authors' reply, and would like to keep my evaluation to this paper.

---

### Official Review · Reviewer_Aht7 · 2026-03-13

**Soundness:** 3
**Presentation:** 3
**Significance:** 3
**Originality:** 3
**Overall Recommendation:** 4
**Confidence:** 3

**Summary:**

This paper proposes PACEAttention that updates features by operating in a low-rank subspace, avoiding both the high computational cost and the overly simplified operations in prior approaches. It shows PACEAttention achieves principled feature updates. The paper argues PACENet displays better interpretability, where different heads attend to distinct fine-grained image regions. PACEAttention also achieves linear complexity in time and memory.

**Compliance With Llm Reviewing Policy:**

Affirmed.

**Final Justification:**

As the rebuttal resolved my main concerns, I maintain my initial positive assessment.

**Key Questions For Authors:**

1. In Eq. (4), why not perform expansion in the subspace while compression not in the subspace, i.e., reverse the two terms?
2. In Section 3.1, how does the similarity between z and bases Uk compute? If Eq.(7) is how they compute, it is unclear why this formulation represents the similarity between z and Uk?
3. Is the basis Uk orthogonormal? How do you enforce that during the neural network training?

4. Why are there 100 layers in the toy experiment? It seems like a big model.
5. What is the training objective for the model pre-training?
6. In Fig.6, why compare PACENet-B (57.7M) with ToST-S (22.6M)? Is it a fair comparison with a different number of parameters?

**Limitations:**

Yes.

**Strengths And Weaknesses:**

### Strengths
1. The paper derives the attention mechanism directly from the geometry of the maximal coding rate reduction objective, providing a clear theoretical motivation for the feature update rules.
2. The expansion and compression operations are formulated through null-space and column-space projections, offering an interpretable mechanism that links network operations with subspace geometry.
3. The proposed approach achieves linear time and memory complexity with respect to the number of tokens, improving scalability compared with prior principled approaches that require quadratic complexity.

### Weaknesses
1. The link between PACE and the attention mechanism is not clear. The attention mechanism is about relations among tokens; it is unclear how the null space projection $I-Q_kQ_k^T$ is treated as attention. In comparison, the distribution of membership, Eq.(7), looks more like an attention map in the attention mechanism.
2. Some of the technical details are not clearly discussed. See questions below.

---

> ### Author Rebuttal · Authors · 2026-03-28
>
> **Weakness1:How the null space projection is treated as attention? Eq.(7) more like attention?**
>
> **Response:** We refer to mechanisms that provide relations among tokens for feature updates as attention. Classical self-attention computes a pairwise "similarity matrix" for feature updates with learnable matrices $W_Q, W_K, W_V$[1,5]. In contrast, MSSA, as a gradient approximation of MCR-like objective, employs a similarity matrix of the form $(U^T Z)^T (U^TZ)$, where $W_Q = W_K = W_V = U$ [4].
>
> In our model, **$Q$ in null space projection can be understood as the low dimensional structure of this similarity matrix.** As shown in Figure 1(b), the gradient of the MCR^2 can be interpreted as performing expansion and compression via null-space projection. This projection is determined by $Q$ and is used for feature updates. Hence, we treat this mechanism as a form of attention.
> Eq.(7) primarily acts as a router, determining which subspace is used for the feature update.
>
> **Q1:Why not reverse expansion and compression terms in Eq.(4)?**
>
> **Response:**
> MCR^2 aims to enlarge the whole feature spaces to separate the different clusters features, and compress feature in the same cluster/subspace[2].
> If the two terms are reversed, it becomes inconsistent with the MCR^2 objective. Moreover, subspace separability primarily relies on the expansion of the overall feature space[3]. Compressing the entire space may lead to feature collapse into a single subspace.
>
> **Q2:how Eq.(7) represents the similarity between z and Uk?**
>
> **Response:** Yes, Eq.(7) is how they compute. Specifically, if a feature does not belong to a given subspace and is orthogonal to it, its projection onto the column space of that subspace is zero. Hence, given a subspace $U_k$, the L2 norm of the column space projection serves as a measure of similarity.
> However, the compression primarily operates on the code of z (i.e., $\alpha_k=U^T_kz$). Therefore, instead of directly measuring the similarity between z and bases $U_k$, we measure the similarity between code of $z$ (i.e., $\alpha_k$) and the codes of all features $U^T_kZ$, where $Q_k$ in Eq.(7) denotes the column space of $U^T_kZ$ (lines 213-215).
>
> **Q3:Is the basis U_k orthonormal? How to enforce it?**
>
> **Response:** Ideally, $U_k$ should be orthonormal. However, enforcing strict orthonormality would require costly manifold optimization (e.g., Stiefel manifold projection). How to efficiently enforce the orthonormality of $U_k$ still remains an open question.
>
> Following prior works[1,4], we did not enforce the orthonormality of $U_k$. In our scheme, the key null-space projection is obtained via Cholesky decomposition(Appendix B.3), where Q is orthonormal. Hence, the resulting core transformation is based on an orthonormalized structure.
>
> Besides, spectral clustering on cosine similarity shows that deeper features exhibit clear subspace structure, indicating that relaxing orthonormality does not significantly affect  subspace representation learning. We will clarify this in the final version.
>
> **Q4:Why 100 layers in toy experiment?**
>
> **Response:**
> The purpose of using 100 layers is to study model stability under an extreme setting. Since the toy dataset is very small, this does not incur significant training cost. Moreover, the model remains lightweight when the number of heads $K$ and feature dimension $d$ are small. As a reference, as shown in Table 3, PACENet-B has 48 layers, yet it contains only 21M parameters. Besides, Figure 13(d) shows the compression results on toy data with 48 layers. A decreasing trend in coding rate similar to that in Figure 3(a) can be observed.
>
> **Q5:Training objective for pre-training?**
>
> **Response:** We follow standard practice and adopt cross-entropy for supervised classification.
> As noted in [1], “the objective function used to derive the architecture and the training objective used to learn the model parameters need not be the same.”
>
> **Q6:In Fig.6, why compare PACENet-B (57.7M) with ToST-S (22.6M)?**
>
> **Response:** As shown in Table 3 (Appendix C.2), **PACENet-B** has 21M parameters, whereas **PACENet-B+** has 57.7M parameters. Hence, the comparison is fair.
>
> **References:**
>
> [1] Wu, Ziyang, et al. "Token statistics transformer: Linear-time attention via variational rate reduction." *arXiv preprint arXiv:2412.17810* (2024).
>
> [2]Chan, Kwan Ho Ryan, et al. "Redunet: A white-box deep network from the principle of maximizing rate reduction." *Journal of machine learning research* 23.114 (2022): 1-103.
>
> [3]Yu, Xiaojie, et al. "ESS-ReduNet: Enhancing Subspace Separability of ReduNet via Dynamic Expansion with Bayesian Inference." *arXiv preprint arXiv:2411.17961* (2024).
>
> [4] Yu, Yaodong, et al. "White-box transformers via sparse rate reduction: Compression is all there is?." *Journal of Machine Learning Research* 25.300 (2024): 1-128.
>
> [5] Vaswani, Ashish, et al. "Attention is all you need." Advances in neural information processing systems 30 (2017).

---

> > ### Author Rebuttal · Reviewer_Aht7 · 2026-04-01
> >
> > Thank you for the response. It has fully addressed my questions

---

> > > ### Author Response · Authors · 2026-04-02
> > >
> > > Thank you for your positive feedback and for confirming that our response has addressed your concerns.

---

### Decision · Program_Chairs · 2026-04-30

**Decision:**

Accept (regular)

**Comment:**

This paper introduces PACEAttention, a novel attention block grounded in the geometric principles of MCR^2 that utilizes randomized subspace projections to achieve feature compression and expansion. By leveraging these projections, the authors develop PACENet, a white-box Transformer architecture that operates with linear complexity relative to the sequence length.

The reviewers agree that the paper is technically solid and presents a clear improvement over previous principled architectures in terms of scalability and performance. The use of randomization to efficiently approximate subspace structures was particularly appreciated as a novel and practical contribution to the field. The strong empirical results on ImageNet and the demonstrated interpretability of the attention heads suggest this work can be of significant interest to the community. Hence, I recommend an acceptance.